# Three diverse motives for information sharing
Valentina Vellani [1,2] ✉, Moshe Glickman [1,2] & Tali Sharot[1,2,3]

Knowledge is distributed over many individuals. Thus, humans are tasked with informing one another for the betterment of all. But as information can alter people's action, affect and cognition in both positive and negative ways, deciding whether to share information can be a particularly difficult problem. Here, we examine how people integrate potentially conflicting consequences of knowledge, to decide whether to inform others. We show that participants (Exp1: $N = 114$, Pre-registered replication: $N = 102$) use their own information-seeking preferences to solve complex information-sharing decisions. In particular, when deciding whether to inform others, participants consider the usefulness of information in directing action, its valence and the receiver's uncertainty level, and integrate these assessments into a calculation of the value of information that explains information sharing decisions. A cluster analysis revealed that participants were clustered into groups based on the different weights they assign to these three factors. Within individuals, the relative influence of each of these factors was stable across information-seeking and information-sharing decisions. These results suggest that people put themselves in a receiver position to determine whether to inform others and can help predict when people will share information.

From financial advisors to doctors and parents—humans are endowed with the task of informing others to aid their decision-making. How do people decide whether to share relevant information? This is a difficult problem to solve, because information can serve several, sometimes competing, goals. Imagine, for example, a teacher who must decide whether to provide a student with negative feedback. The negative feedback may hurt the students' feelings but may be necessary to improve their skills. Thus, the teacher will need to arbitrate between the impact on the student's emotional state (i.e., potential cost) and future performance (i.e., potential benefit), to select a plan of action. The teacher's decision may depend on how much they value (or believe the student values) these different outcomes. Such decisions may be even more complex when the informer is not familiar with the receiver and their preferences and vice versa. This is a common situation in the modern world, where users of online platforms, services, and discussion forums, are often anonymized. Here, we investigate how people solve such complex problems. We hypothesize that people rely on their own *information-seeking* preferences to solve *information-sharing* problems, integrating their preferences over different outcomes into a calculation that leads to information-sharing or its avoidance.

We have recently proposed a theory that characterizes three key motives for information-*seeking*[1]. According to this theory, when deciding whether to seek information, people first estimate what the information will reveal and then estimate the expected impact of that information on their effect (i.e., how the information will make them feel), cognition (how the information will improve their models of the world) and action (how the information will be useful for obtaining rewards). In particular, all else being equal, people will be more likely to seek information (i) when they expect knowledge to make them feel better[2–11], (ii) when uncertainty is high[3,8,10–17], and (iii) when it can aid in selecting action that will help gain rewards and avoid harm[3,18–20].

Different people assign different weights to each of these factors when deciding whether to seek information[3,7]. These are integrated into a computation of the value of information which result in individual differences in information-seeking behavior[3,7]. For example, an individual who prioritizes the affective impact of information, might decide to avoid a medical screening because of fear of bad news, while another who prioritizes the practical use of information in avoiding harm, may attend them religiously.

We hypothesize that people will integrate the three motives above also when deciding whether to share information. That is, people will share information more when it is useful for the receiver, when it may elicit positive emotions in the receiver, and when the receiver is highly uncertain. Different people may assign different weights to these motives, which will account for

[1]Affective Brain Lab, Department of Experimental Psychology, University College London, London, WC1H 0AP, UK. [2]Max Planck University College London Centre for Computational Psychiatry and Ageing Research, London, WC1B 5EH, UK. [3]Department of Brain and Cognitive Sciences, Massachusetts Institute of Technology, Cambridge, MA, USA. ✉e-mail: vellaniuni@gmail.com

different patterns of information-sharing. While there are clues in the literature that when informing other people also prefer to *share* information that is positive[19,21–33] and that can guide action[21,22,34], more studies are needed to computationally disentangle the importance of these (sometimes) competing motives. Rather, motives have either been studied in isolation or in a situation where they are confounded. Moreover, whether people consider the receiver's level of uncertainty in making sharing decisions is unknown. In real life, conflicting outcomes of knowledge for the receiver are often present. Thus, characterizing information-sharing in such situations is crucial for understanding how people decide whether to inform others.

Here, we *simultaneously* varied the instrumental utility of information that could be shared, the level of uncertainty of the receiver, and the valence of information (Exp1: *N* = 114; Preregistered Replication: *N* = 102). We then examine how these considerations are integrated into a sharing decision and whether sharers weigh these factors as they do when they themselves make information-seeking decisions. In contrast to information-seeking, when people share information, they often know the content of that information. Thus, if information-seeking resembles information-sharing, that may indicate that the sharer considers the point of view of the receiver in respect to the uncertainty they are under (which is different from their own), as well as considering its potential impact on the receiver's affect and the instrumentally for the receiver.

## Methods
### Participants
Participants were recruited via Prolific Academic (https://www.prolific.co/) and were paid £7.50 per hour for their participation. The study was approved by the departmental ethics committee at UCL. Informed consent was obtained by all participants. Ethnicity data has not been collected. All demographic information including gender has been self-reported by participants. All tasks were created using Gorilla Experiment Builder[35] (www.gorilla.sc). The sample size was determined based on our previous study on information-seeking[7].

In Exp1, 124 participants performed an information-seeking task of which one participant was excluded for failing all catch trials (see procedure for more details) and one because of an error in data storing. The final sample was composed of 122 participants (55 males, 63 females, and 4 others, mean age = 31.42 years ±9.7 (SD), age range: 18–60 years). In total, 128 participants performed an information-sharing task of which two participants were excluded for failing all catch trials and one for completing the task twice. The final sample was composed of 125 participants (54 males, 71 females, mean age = 32.28 years ±9.09 (SD), age range: 18–60 years). Out of all participants, 56 participants replied to our invitation to participate in the study again (to complete either the seeking or the sharing task) and therefore completed both the seeking and sharing task. One participant completed the information-sharing task twice so their data was not included, therefore data from 55 participants was analyzed.

To estimate the required sample size for the replication, we conducted a power analysis on the weakest effect observed (trend found in the Robust Regression analysis (Huber) testing whether participants who assigned greater weight to instrumentality when seeking information themselves also assigned greater weight to it when deciding whether to share information). The power analysis (power = 0.8, alpha = 0.05) revealed a required sample of 119 subjects. We collected data from 122 participants to account for dropouts (53 males, 67 females, 3 others, mean age = 35.8 years ±10.2 (SD), age range: 18–60 years). They completed both the information-seeking and sharing tasks. In the seeking task, data from eleven participants was excluded from the analysis as they failed all catch trials. Data from one participant was excluded from the analysis as they completed the task twice. In the sharing task, data from five participants was excluded from the analysis as they failed all catch trials. Thus, the final sample was composed of 110 participants for the seeking task and 117 for the sharing task. Of these subjects, 108 completed both tasks. The replication was preregistered https://osf.io/ecxsr/?view_only=bd9c51b1d58149ab9901dfd842d33dc8 on September 20th, 2022 on September 20th, 2022.

### Procedure
**Information-seeking task**. Following instructions, participants answered six comprehension questions before the first block and one question before the second block. Participants who responded incorrectly twice on at least one question were not permitted to go on to complete the task. After reading the instructions, participants completed two example trials before starting the actual task.

The task is illustrated in Fig. 1a. Recipients were told they owned 100 stocks in a financial market we created. On each trial recipients were presented with an algorithm's prediction of the value of their stocks and the algorithm's average prediction accuracy (the algorithms could be different on each trial). These cues were presented for 5 s. Predictions regarding the stocks' value ranged from −400 to −500 and from +400 to +500. A positive stock value meant recipients were earning money, a negative value meant they were losing money. We use the word 'valence' to indicate both the direction of the change in stock's value (positive or negative) and the magnitude of such change. The algorithm's prediction accuracy ranged from 0 to 99%, high numbers suggest the algorithm is often correct and vice versa. The algorithm's accuracy represents the probability of the revealed stocks' value being the real value. When the prediction is incorrect, the stocks' value is uniformly distributed over the possible values. Stocks' value and uncertainty level were randomly sampled for each trial. Recipients then indicated whether they wanted to open an envelope containing information about the true value of their stocks (information-seeking decision). They did so using a seven-point Likert scale ranging from 0, "Not at all," to 6, "Very much". We adopted a 7-point Likert scale rather than a binary response scale to increase the sensitivity of our measure. Recipients were told that the closer their answer was to "Very much," the more likely we were to open the envelope and reveal the value of their stocks, and vice versa (similar to other studies[2,3,36]. If they selected 0, 1, 2, 3, 4, 5, or 6, then the information was delivered with a probability of 5%, 20%, 35%, 50%, 65%, 80%, and 95%, respectively. This probabilistic approach, which is adapted from past studies[2,3] is reflective of real life where actions are often not deterministically related to outcomes (e.g., one may ask a question but not receive an answer and may need to ask again if they really want to know). Recipients were not aware of these exact mathematical conversions as we did not want them to focus their attention on exact mathematical calculations to avoid distraction. Next, either the value of their stocks (information) was presented on screen for 4 s or hidden ('XX' was shown).

The task was composed of two blocks. In the instrumental block, recipients were informed that on each trial, they would be able to decide whether to add 10 stocks to their portfolio, give away 10 stocks, or leave the number of their stocks as is (financial decision). In the non-instrumental block, recipients were informed that the computer would randomly make this decision for them. Recipients were informed that they would start the task with 250 K bonus points which were worth between £1 and £5 total. This ambiguity was added to reflect real life, where the exact material value of choices is often uncertain. At the end of the task, the Gorilla program randomly selected one trial, and the value of their stocks on that trial was multiplied by the number of stocks they had. The resulting sum was added to their initial bonus. For example, if, on the selected trial, they had 200 stocks worth −450 points, 90 K (450*200) points would be subtracted from their initial bonus of 250 K. The order of blocks was counterbalanced across individuals. Each block was composed of 44 trials plus 4 catch trials.

Catch trials were added to check participants' engagement and attention. In those trials, instead of indicating how much they wanted to share/seek information, participants were instructed to select a specific rating (for example: *Select 1*). Participants who failed all catch trials in one of the blocks were excluded from the analysis.

In addition, to check whether participants were encoding the information provided, on four trials (memory check trials) in each block, we asked participants to recollect whether the algorithm predicted the stocks to be positive or negative, and/or we asked them what the predicted accuracy of the algorithm was. Those trials were excluded from the analysis, therefore 40 trials for each block were analyzed. Results indicated good attention and

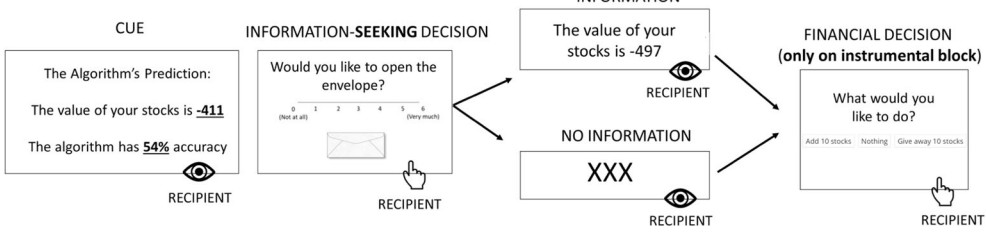

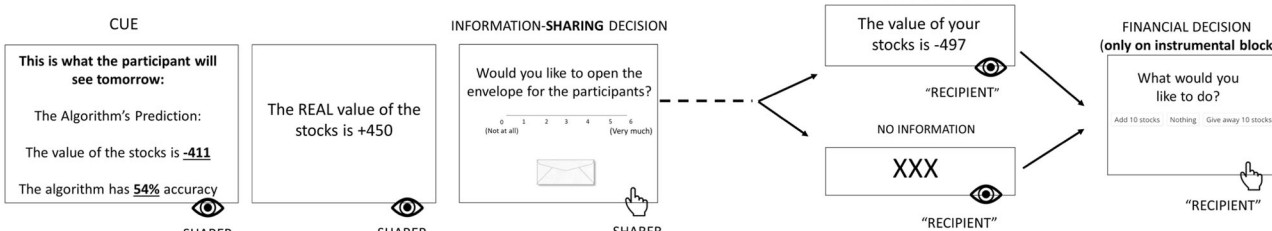

**Fig. 1 | Task.** In the information seeking task recipients were given stocks in a financial market we created. On each trial they observed an algorithm's prediction of the current value of their stocks and the prediction accuracy of the algorithm (**cue**). Participants then indicated on a scale from 0 (not at all) to 6 (very much) whether they wanted to open an envelope to observe the current value of their stocks (**informing self**). If they indicated they wanted to open the envelope they were likely to then observe the current value of their stocks (**information**). If they indicated they did not want to open the envelope they were then more likely to observe 'XX' (**no-information**). In the instrumental block recipients could then decide whether to add 10 stocks, give away 10 stocks or leave the number of stocks as is (**financial decision**). On the non-instrumental block, a computer randomly made the decision for them. In the information sharing task, on each trial sharers first observed an algorithm's prediction about the value of stocks of a "recipient" that may be playing a similar task tomorrow and the prediction accuracy of the algorithm (**cue**). Then they observed the actual value of the stocks. Next, they indicated whether they wanted to open the envelope for tomorrow's "recipient" so that the "recipient" could observe the value of their stocks on that trial (**informing others**). Sharers were told that if they indicated they wanted to open the envelope for the "recipient", the "recipient" was then more likely to observe the value of their stocks (**information**). If they indicated they did not want to open the envelope for the "recipient", the "recipient" was then more likely to observe 'XX' (**no-information**). On the instrumental block of trials sharers were told the "recipient" could then decide whether to add or give away stocks ("recipient's financial decision). In the non-instrumental block, sharers were told the "recipient" was not able to make that decision. In reality, there were no participants playing the next day. Thus, parts of the trial marked as "recipient" in the sharing task represent only what the participants believed would happen the next day. Post-task questionnaires suggest participants believed they were interacting with another participant and were trying to help others (see Supplementary Results). There were 2 blocks, each composed of 40 trials.

memory—on average, 87.5% of participants in Exp1 and 90.4% of participants in the replication provided the correct response.

**Information-sharing task.** The information-sharing task (Fig. 1b) was nearly identical to the information-seeking task described above. The difference was that in the sharing task, participants (sharers) did not own stocks. Rather, they were told "recipients", who may play the task tomorrow, will own stocks in the market. On each trial, they were presented with the algorithm's predictions regarding the stocks of those "recipients" and the prediction accuracy of that algorithm (cue). They were aware the "recipient" tomorrow would also observe these cues. Then they received an open envelope containing the actual value of the recipients' stocks. They were then asked to indicate whether they wanted to share the envelope's content with tomorrow's "recipients" so that they could observe the value of their stocks on that trial (information-sharing decision) on a 7-point Likert scale ranging from 0 "Not at all" to 6 "Very much". In the instrumental block sharers were informed that tomorrow's "recipients" would be able to use the information provided to decide whether to add or give away stocks (other's financial decision). Sharers were told that the closer their answer was to "Very much", the greater the probability that we will open the envelope and reveal the value of the stocks, and vice versa (similar to the seeking task). In the non-instrumental block sharers were informed tomorrow's "recipients" could not use the information shared with them. In this task, there was no monetary incentive to share information with the other players. Analysis of the debriefing questionnaire revealed many participants shared information because they were trying to help others (see Supplementary

results). The order of blocks was counterbalanced across subjects. Results indicated good attention and memory—on average, 89.1% of participants in Exp1 and 88.03% in the replication provided the correct response to the memory questions. Analysis of the debriefing questionnaire revealed participants believed they were interacting with another participant (see Supplementary results).

**Additional Information.** Out of the final sample, 55 participants in Exp1 and all 108 participants in the replication study performed both the information-seeking and sharing tasks. The order of the tasks was counterbalanced. For these 55 participants, we also included a set of personality questionnaires (The Apathy Evaluation Scale (AES), The State-Trait Anxiety Inventory (STAI), the Revised Life Orientation Test (LOT-R), Zung Self-Rating Depression Scale (SDS)). These served as a pilot for potential future studies. There was no statistically significant evidence that these scores were associated with weights assigned to different factors when sharing or seeking information after correcting for multiple comparisons, though we note sample size is too small to detect such associations.

**Post-task survey**

Sixty-seven participants who completed the sharing task in Exp1 at the end of the study were asked ("How did you decide whether to open the envelope for the other participants?"). Two naïve observers independently rate all responses to indicate whether the responder:

1. believed they were interacting with another participant. (Options: (a) the responder believes they are interacting with another participant; (b)

the responder does not believe they are interacting with another participant; (b) no indication either way).

2. is trying to help the other participant. (Options: (a) the responder is trying to help the other participant;

(b) the responder is not trying to help the other participant; (c) no indication either way).

The observers' answers were entered into an Inter Correlation Coefficient analysis to assess the agreement between them.

## Statistical analysis

**Model estimation.** For each participant, two linear Regressions models were run to predict information sharing and information seeking on each trial, from the level of *uncertainty* (calculated by subtracting the algorithm's accuracy percentage from 100), *valence* of information (valence was defined according to the signed algorithm' prediction value in the information-seeking task, and as the actual signed value of the stocks in the information-sharing task) and *instrumentality* (whether the information could be used to make decisions to alter the portfolio or not). All variables were *z*-scored to obtain standardized betas. Betas across participants were compared to zero using a one-sample *t*-test (all tests are always two-sided). There was no statistically significant evidence that valence, instrumentality and uncertainty were correlated with each other (see Supplementary Table S4). Correlation coefficients were compared against zero with One-Sample t-tests. The model coefficients could not be estimated for subjects with insufficient variability in their sharing or seeking decisions (that is, all decisions were identical). These participants were, therefore, excluded from the analyses. As a consequence, the model was estimated for 109 (Exp1) and 100 (Replication) subjects in the seeking task and for 114 (Exp1) and 102 (Replication) subjects in the sharing task.

Normality assumption was made for all *t*-tests and regression analyses based on the central limit theorem, given the large number of participants (Exp1 $N = 114$, Rep $N = 102$) in the data.

**Cluster analysis.** From the previous analysis, we obtained three beta coefficients (instrumentality, valence, and uncertainty) for each participant. These beta values were then used in k-means cluster analyses[37], which reveals groups of participants who show similar patterns of motive prioritization. To determine the optimal number of clusters in the information sharing and seeking conditions in Exp1 and in the replication experiment, we varied $K$ between 2 to 7 and evaluated the quality of clustering for each $K$, using the Calinski–Harabasz index[38]. Across datasets and measures, $K = 3$ consistently yielded the optimal clustering.

We characterized each cluster by averaging the beta coefficients (instrumentality, valence, and uncertainty) across its participants. Using one-sample t-tests, we then compared each cluster's betas to zero. Next, we examined whether people who belong to one cluster when seeking information tended to belong to the same cluster when sharing it. A binomial test was used to test whether this consistency was significantly greater than chance (33.3%).

Finally, we examined the correlations between the beta coefficients of uncertainty, valence, and instrumentality across the information-sharing and seeking tasks for participants who completed both tasks using Robust Regression analysis (Huber).

## Results

In this study, we examine how instrumental utility, level of uncertainty, and the valence of information are integrated into information sharing and seeking decisions and whether people weigh these factors while sharing information as they do when they seek information. To assess whether individual differences are stable across informing self and informing others' decisions, we tested whether weights assigned to each factor when seeking information are correlated to its weight when sharing.

To that end, we conducted two experiments. In Exp1, participants ($N = 114$) performed an information-sharing task ('sharers', Fig. 1), an information-seeking task ($N = 109$), or both ($N = 55$ out of the numbers above). In both tasks, we manipulated (i) the valence of information for the recipient, (ii) the level of uncertainty of the recipient, and (iii) the instrumental utility of information for the recipient, as described in detail below. We then pre-registered a replication study (https://osf.io/ecxsr/?view_only=bd9c51b1d58149ab9901dfd842d33dc8) where participants ($N = 108$) completed both tasks ($N = 11$ did only one task due to failing attention checks). As indicated below, all findings replicate across the two studies.

## Participants consider the impact of information on affect, action, and uncertainty when deciding whether to inform others

We tested whether participants considered the valence of information, the receiver's uncertainty, and the instrumentality of information when making information-sharing and seeking decisions. A linear regression was run for each individual to predict separately (a) information-sharing and (b) information-seeking, on each trial from three factors: (i) level of *un*certainty (equal to 100 minus the algorithms' accuracy), (ii) instrumentality (coded as 1 if the information could be used to alter the portfolio and 0 otherwise) and (iii) valence of information (the stocks' value ranged from −400 to −500 and from +400 to +500. As a reminder—valence, uncertainty, and instrumentality are not correlated, see Supplementary Table S4. In the information-seeking task, the stock's value was based on the algorithm's prediction, while in the information-sharing task, it was based on the actual stock value. The obtained betas are then compared to zero using a *t*-test (Table 1). the model was estimated for 109 (Exp1) and 100 (Replication) subjects in the seeking task and for 114 (Exp1) and 102 (Replication) subjects in the sharing task (see "Statistical analysis").

$$\text{Information decision} = \beta 0 + \beta 1 * \text{Uncertainty} + \beta 2 * \text{Instrumentality} + \beta 3 * \text{Valence}$$

As observed in Table 1 below, participants were more likely to share and seek information when (i) the receiver's uncertainty was high, (ii) when information was instrumental to the receiver, and (iii) when the information would likely convey good news—that is when the expected/true value of the stocks was high. The median $R^2$ scores across individuals were as follows for information-seeking: Exp1: median = 0.51, SE: 0.03, Rep: median = 0.46, SE: 0.03, and for sharing: Exp1: median = 0.37, SE: 0.02, Rep: median = 0.40, SE: 0.03.

We then investigated the difference between the weights assigned to each beta in the sharing and seeking tasks. To do so we run independent samples *t*-tests between weight assigned to each beta in the sharing and seeking tasks (considering both Exp1 and Rep together). Results showed that the instrumental value of information was higher when seeking ($M = 0.41$) than when sharing information ($M = 0.28$; $t(132) = 3.36$, $p = 0.001$), indicating that participants weighted the instrumental value of information more when seeking information than sharing it.

The results remain exactly the same when adding the trial number as a covariate to the linear regressions to control for fatigue. The results also remain the same when controlling for the two-way and three-way interactions between the motives. We performed an exploratory analysis of these interactions, and the results are reported in the Supplementary Results section (Table S3). In addition, the same pattern of results was obtained by running two separate linear mixed-effects models to predict information-sharing and information-seeking on each trial from uncertainty, instrumentality, and (iii) valence (see details in Supplementary Results—Table S1 S2 and Fig. S1).

Figure 2 presents the raw data of information seeking and sharing across all participants and trials in both Experiment 1 and the replication as a function of instrumentality, uncertainty, and valence. The figure illustrates positive relationships between information seeking/sharing and these three factors in both experiments.

**Table 1 | Average Beta coefficient for predicting information seeking and sharing**

| Exp1 | Motives | M | 95% CI | t | df | p |
|---|---|---|---|---|---|---|
| Information-sharing | Uncertainty | 0.09 | [0.01, 0.16] | 2.36 | 113 | 0.02 |
| Information-sharing | Instrumentality | 0.21 | [0.14, 0.28] | 6 | 113 | <0.001 |
| Information-sharing | Valence | 0.10 | [0.05, 0.15] | 4.64 | 113 | <0.001 |
| Information-seeking | Uncertainty | 0.14 | [0.07, 0.20] | 4.28 | 108 | <0.001 |
| Information-seeking | Instrumentality | 0.39 | [0.31, 0.46] | 10.14 | 108 | <0.001 |
| Information-seeking | Valence | 0.11 | [0.06, 0.15] | 4.96 | 108 | <0.001 |
| **Replication** | **Motives** | **M** | **95% CI** | **t** | **df** | **p** |
| Information-sharing | Uncertainty | 0.18 | [0.11, 0.26] | 4.79 | 101 | <0.001 |
| Information-sharing | Instrumentality | 0.26 | [0.18, 0.33] | 6.76 | 101 | <0.001 |
| Information-sharing | Valence | 0.06 | [0.03, 0.09] | 3.71 | 101 | <0.001 |
| Information-seeking | Uncertainty | 0.19 | [0.10, 0.27] | 4.47 | 99 | <0.001 |
| Information-seeking | Instrumentality | 0.33 | [0.25, 0.41] | 8.41 | 99 | <0.001 |
| Information-seeking | Valence | 0.06 | [0.02, 0.09] | 3.15 | 99 | 0.002 |

Participants prefer to share and seek information when (i) the receiver's uncertainty is high, (ii) the information is instrumental to the receiver, and (iii) the information would likely convey good news—that is when the expected/true value of the stocks was high. The table shows the average Beta coefficient from linear regression across individuals that predict separately (a) information-sharing and (b) information-seeking on each trial from three factors: (i) level of uncertainty, (ii) instrumentality-, and (iii) valence of information. p and t are from a t-test against zero.

**Fig. 2 | Raw data of information seeking ($N_{\text{Seeking}} = 114$) and sharing ($N_{\text{Sharing}} = 109$) across participants and trials in Exp. 1 (top two rows) and Replication (bottom two rows; $N_{\text{Seeking}} = 102$ and $N_{\text{Sharing}} = 100$).** The figure shows mean information-seeking/sharing decisions (y-axis) as a function of uncertainty (left column, binned in groups of 10), instrumentality (middle column), and valence (right column, binned as negative/positive). Black lines represent mean decisions across participants. Gray dots represent individual data points for each participant and trial. Higher values indicate a greater willingness to seek/share information.

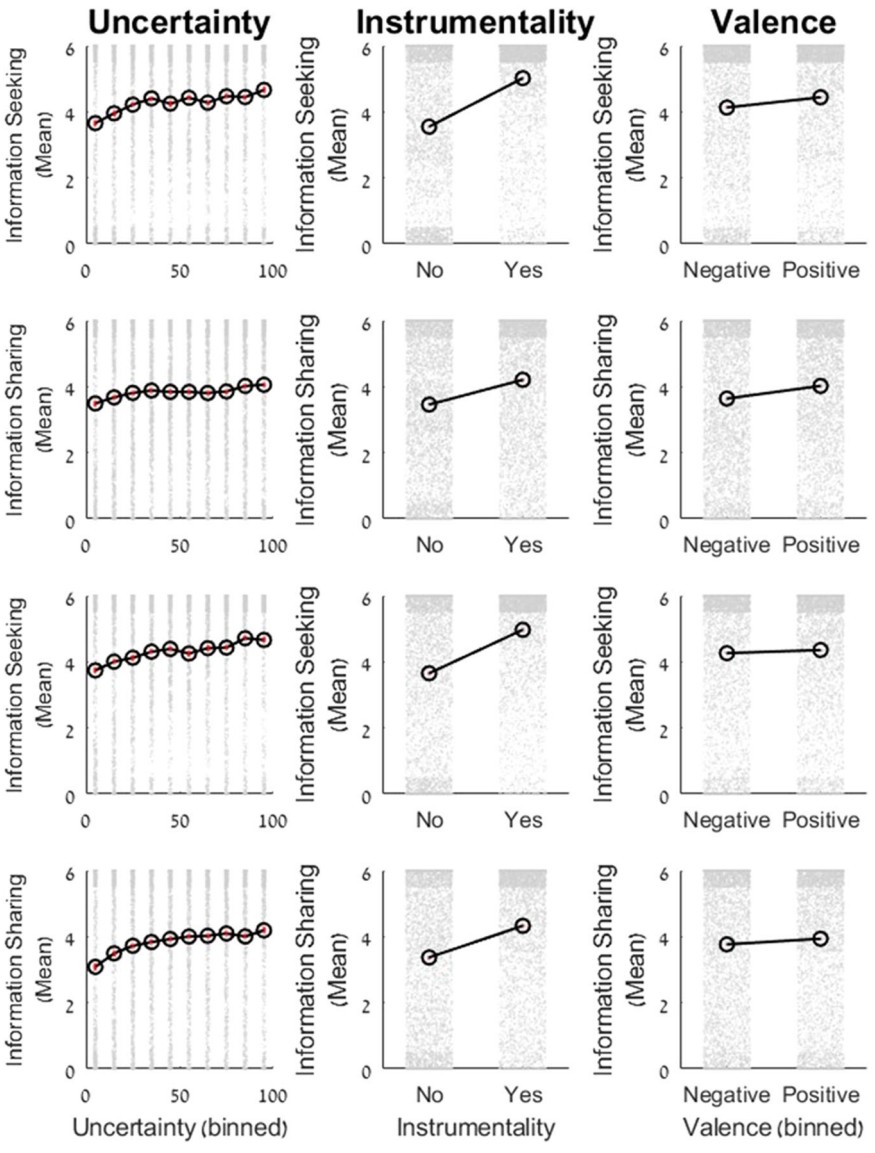

## Individual differences in sharing decisions

The results thus far suggest that when deciding whether to inform others, people consider the receiver's uncertainty, the instrumental utility of information, and its valence. However, it is likely that different people put different weights on these factors. Indeed, it has been shown individuals often weigh one of these factors over and above the others[7]. As a result, information-seeking decisions are vastly different across individuals. Here, we tested whether similar individual differences are observed when sharing information. To that end, we entered participants' individual beta coefficients from the linear regressions detailed above into a K-means cluster analysis separately for information-seeking decisions and information-sharing decisions. To determine the optimal number of clusters, we calculated the Calinski–Harabasz index[38] (see details in the "Methods" section, Table 2). Across datasets and measures, $K = 3$ consistently yielded the optimal clustering solution. For each cluster, betas have been tested against zero using a one-sample $t$-test.

**Table 2 | The Calinski–Harabasz index for each experiment (Exp. 1 — Info Seeking, Exp. 1 — Info Sharing, and Replication - Info Seeking and Sharing) as a function of K**

| Calinski–Harabasz index | | | | | | |
|---|---|---|---|---|---|---|
| K | 2 | 3 | 4 | 5 | 6 | 7 |
| Exp1—Seeking | 72.36 | **93.17** | 87.88 | 81.74 | 81.32 | 78.15 |
| Exp1—Sharing | 61.14 | **72.90** | 69.80 | 71.48 | 70.29 | 68.77 |
| Replication—Seeking | 56.83 | **98.01** | 88.78 | 84.13 | 83.97 | 80.36 |
| Replication—Sharing | 59.73 | **86.56** | 77.45 | 68.37 | 65.91 | 63.28 |

Across datasets and measures, $K = 3$ yielded the optimal clustering solution.

Cluster analysis of information-seeking decisions revealed the following three distinct groups of participants across experiments and conditions (Table 3, Fig. 3b, d, f, h). The first cluster, labeled "Uncertainty-Dominant Group", included 22% of participants in Exp1 and 34% in the Replication study who assigned a large positive weight to uncertainty, a moderate positive weight to instrumentality and no significant weight to valence when seeking information.

The second cluster, labeled "Instrumentality-Dominant Group", included 39.44% of participants in Exp1 and 35% of participants in the Replication study who assigned a large positive weight to instrumentality when seeking information, with weights on valence and uncertainty being moderate positive or not significant.

The third cluster, labeled "Valence-Dominant Group" (Exp1) "Valence-Certainty Dominant Group" (Replication), included 38.53% of participants in Exp1 and 31% in the Replication study who assigned a large positive weight to valence when seeking information, with weight on instrumentality being moderate positive or non-significant and either non-significant or negative on uncertainty (thus positive on certainty).

The analysis on sharing decisions revealed a similar pattern by which participants clustered into the following three groups (Table 4, Fig. 3 a, c, e, g). The first cluster ("Uncertainty-Dominant Group"), included 37.71% of participants in Exp1 and 48.03% of participants in the Replication study who assigned a large positive weight to uncertainty when sharing information, with no significant weight on valence.

The second cluster, which we will call the "Instrumentality-Dominant Group", included 28.07% of participants in Exp1 and 32.35% of participants in the Replication study who assigned a large positive weight to instrumentality when sharing information, and a moderate weight on and uncertainty.

The third cluster, which we will call the "Valence- Certainty-Dominant Group", included 34.21% of participants in Exp1 and 19.60% of participants in the Replication study who assigned a large positive weight to valence and a

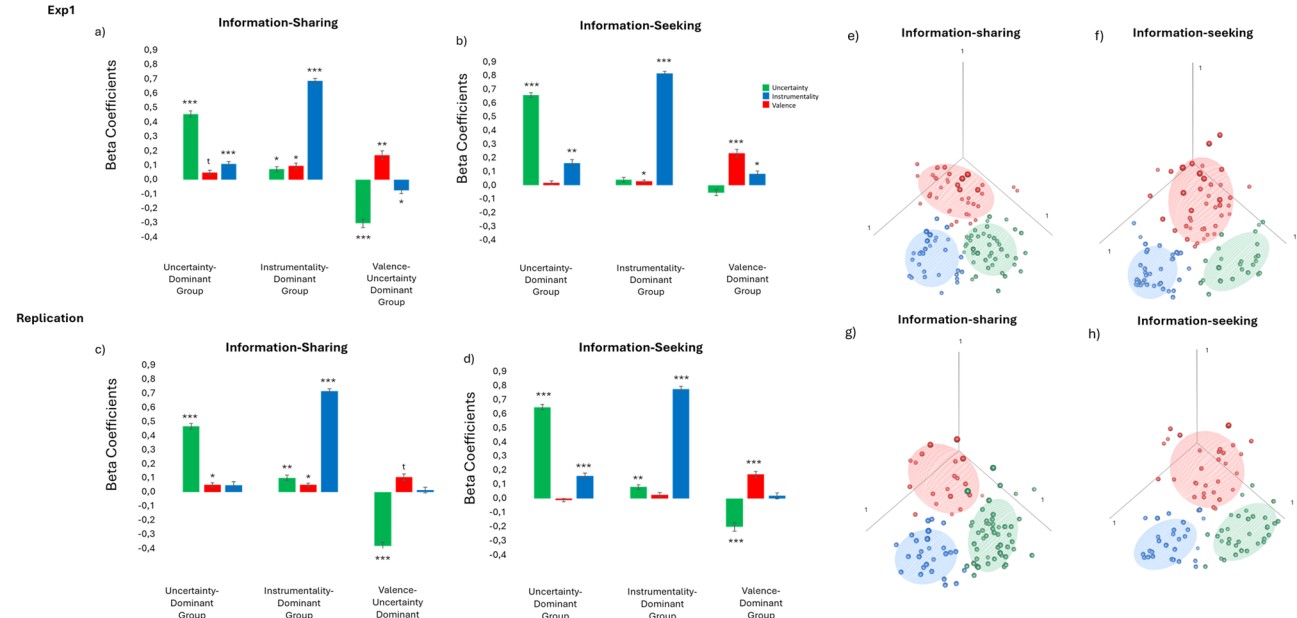

**Fig. 3 | Participants cluster into three groups, characterized by the weight they assign to the valence of information, its instrumentality, and the receiver's uncertainty when deciding to inform the self and others.** We calculated the weights each participant assigned to each of the three factors (instrumentality, valence, and uncertainty) when seeking and sharing information (Exp. 1: $N_{Seeking} = 114$ and $N_{Sharing} = 109$; Replication: $N_{Seeking} = 102$ and $N_{Sharing} = 100$). The obtained betas were submitted into a cluster analysis to identify groups of participants that have similar combinations of weights when seeking or sharing information. **a–d** Plotted are the average beta coefficients assigned to each factor, averaged across participants in each cluster. As can be seen, the Instrumentality-

Dominant group put the most weight on the instrumental value of information, the valence–dominant group put the most weight on valence, the Uncertainty-Dominant group put the most weight on uncertainty, and the Valence–Certainty–Dominant group put the most weight on valence and negative weight on uncertainty (which can be framed as positive weights on certainty). **e–h** The weights of individual participants assigned to each of the three motives are plotted with participants colored according to their assigned cluster. Ellipsoid highlights 50% of the data. * $p < 0.05$, ** $p < 0.01$, *** $p < 0.001$; Error bars represents SEM.

**Table 3 | Cluster analysis on information-seeking**

| Exp1 | | | | | | Replication | | | | | |
|---|---|---|---|---|---|---|---|---|---|---|---|
| **Uncertainty-Dominant Group** | | | | | | **Uncertainty-Dominant Group** | | | | | |
| **Motives** | **Mean β** | **95% CI** | **t** | **df** | **p** | **Motives** | **Mean β** | **95% CI** | **t** | **df** | **p** |
| Uncertainty | 0.658 | [0.58, 0.72] | 19.15 | 23 | <0.001 | Uncertainty | 0.648 | [0.58, 0.71] | 20.48 | 33 | <0.001 |
| Valence | 0.018 | [−0.03, 0.07] | 0.66 | 23 | 0.51 | Valence | −0.013 | [−0.5, 0.02] | −0.65 | 33 | 0.51 |
| Instrumentality | 0.163 | [0.05, 0.27] | 3.16 | 23 | 0.004 | Instrumentality | 0.158 | [0.08, 0.23] | 4.20 | 33 | <0.001 |
| **Instrumentality-Dominant Group** | | | | | | **Instrumentality-Dominant Group** | | | | | |
| **Motives** | **Mean β** | **95% C** | **t** | **df** | **p** | **Motives** | **Mean β** | **95% CI** | **t** | **df** | **p** |
| Uncertainty | 0.040 | [−0.01, 0.09] | 1.42 | 42 | 0.16 | Uncertainty | 0.081 | [0.02, 0.14] | 2.88 | 34 | 0.006 |
| Valence | 0.028 | [0.001, 0.05] | 2.16 | 42 | 0.036 | Valence | 0.026 | [−0.02, 0.08] | 0.96 | 34 | 0.34 |
| Instrumentality | 0.817 | [0.76, 0.86] | 33.73 | 42 | <0.001 | Instrumentality | 0.775 | [0.70, 0.84] | 23.31 | 34 | <0.001 |
| **Valence-Dominant Group** | | | | | | **Valence-Certainty-Dominant Group** | | | | | |
| **Motives** | **Mean β** | **95% CI** | **t** | **df** | **p** | **Motives** | **Mean β** | **95% CI** | **t** | **df** | **p** |
| Uncertainty | −0.053 | [−0.12, 0.01] | −1.55 | 41 | 0.128 | Uncertainty | −0.201 | [−0.31, −0.09] | −3.73 | 30 | <0.001 |
| Valence | 0.234 | [0.14, 0.32] | 5.22 | 41 | <0.001 | Valence | 0.170 | [0.09, 0.24] | 4.50 | 30 | <0.001 |
| Instrumentality | 0.084 | [0.01, 0.15] | 2.51 | 41 | 0.016 | Instrumentality | 0.018 | [−0.06, 0.09] | 0.46 | 30 | 0.64 |

Decisions revealed that participants were clustered into the following three groups: (i) "Uncertainty-Dominant Group", "Instrumentality-Dominant Group" and "Valence-Dominant Group"(Exp1)/"Valence-Certainty-Dominant Group"(Replication)/. For each cluster a one-sample *t*-test was performed for each Beta.

**Table 4 | Cluster analysis on information-sharing**

| Exp1 | | | | | | Replication | | | | | |
|---|---|---|---|---|---|---|---|---|---|---|---|
| **Uncertainty-Dominant Group** | | | | | | **Uncertainty-Dominant Group** | | | | | |
| **Motives** | **Mean β** | **95% CI** | **t** | **df** | **p** | **Motives** | **Mean β** | **95% CI** | **t** | **df** | **p** |
| Uncertainty | 0.457 | [0.39, 0.52] | 13.94 | 42 | <0.001 | Uncertainty | 0.466 | [0.41, 0.52] | 16.75 | 48 | <0.001 |
| Valence | 0.049 | [−0.003, 0.10] | 1.89 | 42 | 0.06 | Valence | 0.050 | [0.006, 0.09] | 2.28 | 48 | 0.026 |
| Instrumentality | 0.11 | [0.05, 0.16] | 3.83 | 42 | <0.001 | Instrumentality | 0.049 | [−0.01, 0.11] | 1.45 | 48 | 0.15 |
| **Instrumentality-Dominant Group** | | | | | | **Instrumentality-Dominant Group** | | | | | |
| **Motives** | **Mean β** | **95% CI** | **t** | **df** | **p** | **Motives** | **Mean β** | **95% CI** | **t** | **df** | **p** |
| Uncertainty | 0.073 | [0.005, 0.14] | 2.19 | 31 | 0.03 | Uncertainty | 0.100 | [0.03, 0.17] | 3.01 | 32 | 0.004 |
| Valence | 0.096 | [0.02, 0.17] | 2.59 | 31 | 0.01 | Valence | 0.050 | [0.007, 0.09] | 2.38 | 32 | 0.02 |
| Instrumentality | 0.685 | [0.61, 0.75] | 19.39 | 31 | <0.001 | Instrumentality | 0.714 | [0.64, 0.78] | 20.64 | 32 | <0.001 |
| **Valence-Dominant Group** | | | | | | **Valence-Certainty-Dominant Group** | | | | | |
| **Motives** | **Mean β** | **95% CI** | **t** | **df** | **p** | **Motives** | **Mean β** | **95% CI** | **t** | **df** | **p** |
| Uncertainty | −0.303 | [−0.40, -0.20] | −6.21 | 38 | <0.001 | Uncertainty | −0.380 | [−0.31, −0.09] | −6.46 | 19 | <0.001 |
| Valence | 0.171 | [0.07, 0.27] | 3.47 | 38 | 0.001 | Valence | 0.104 | [0.09, 0.24] | 1.92 | 19 | 0.06 |
| Instrumentality | −0.074 | [−0.001, −0.14] | −2.07 | 38 | 0.044 | Instrumentality | 0.015 | [−0.06, 0.09] | 0.34 | 19 | 0.73 |

Decisions revealed that participants were clustered into the following three groups "Uncertainty-Dominant Group", "Instrumentality-Dominant Group" and "Valence-Certainty-Dominant Group". For each cluster a one-sample *t*-test was performed for each beta.

negative weight on uncertainty when sharing information. In other words, these participants preferred to share information when the receiver was more certain that the stocks' value was positive.

Note, that the cluster analysis groups together participants who share similar beta patterns. While the analysis yielded three distinct clusters, these do not correspond directly to the three motives (uncertainty, instrumentality, and valence), but rather reflect more complex natural patterns in the data.

**Individual differences are stable across information-seeking and information-sharing decisions**
We next examined whether, within each individual, the weight participants assign to the different outcomes of information when deciding whether to inform others is correlated with the weight they assign to these factors when deciding whether to inform the self (i.e., information-sharing). 55 and 108

participants, respectively, in Exp1 and in the replication, completed both the information-seeking and information-sharing tasks in random order. Of these participants, the model coefficient could be estimated (sufficient variability in the sharing and seeking decisions) for 43 participants for Exp1 and 90 for the Replication. Robust Regression analysis (Huber) revealed that participants who assigned greater weight to a particular factor when seeking information themselves also assigned greater weight to that factor when deciding whether to share information (Exp. 1: Uncertainty: $b = 0.66$, $r = 0.75$, $t(41) = 7.38$, $p < 0.001$, 95% CI [0.47, 0.84], Fig. 4a; Valence: $b = 0.19$, $r = 0.47$, $t(41) = 2.72$, $p = 0.009$, 95% CI [0.05, 0.33], Fig. 4b; Instrumentality: $b = 0.15$, $r = 0.18$, $t(41) = 1.13$, $p = 0.26$, 95% CI [−0.12, 0.42], Fig. 4c; Replication: Uncertainty $b = 0.83$, $r = 0.75$, $t(88) = 10.52$, $p < 0.001$, 95% CI [0.68, 0.99], Fig. 4d; Valence: $b = 0.41$, $r = 0.41$, $t(88) = 4.21$, $p < 0.001$, 95% CI [0.21, 0.60], Fig. 4e; Instrumentality: $b = 0.47$, $r = 0.43$, $t(88) = 4.47$, $p < 0.001$, 95% CI [0.26, 0.68], Fig. 4f).

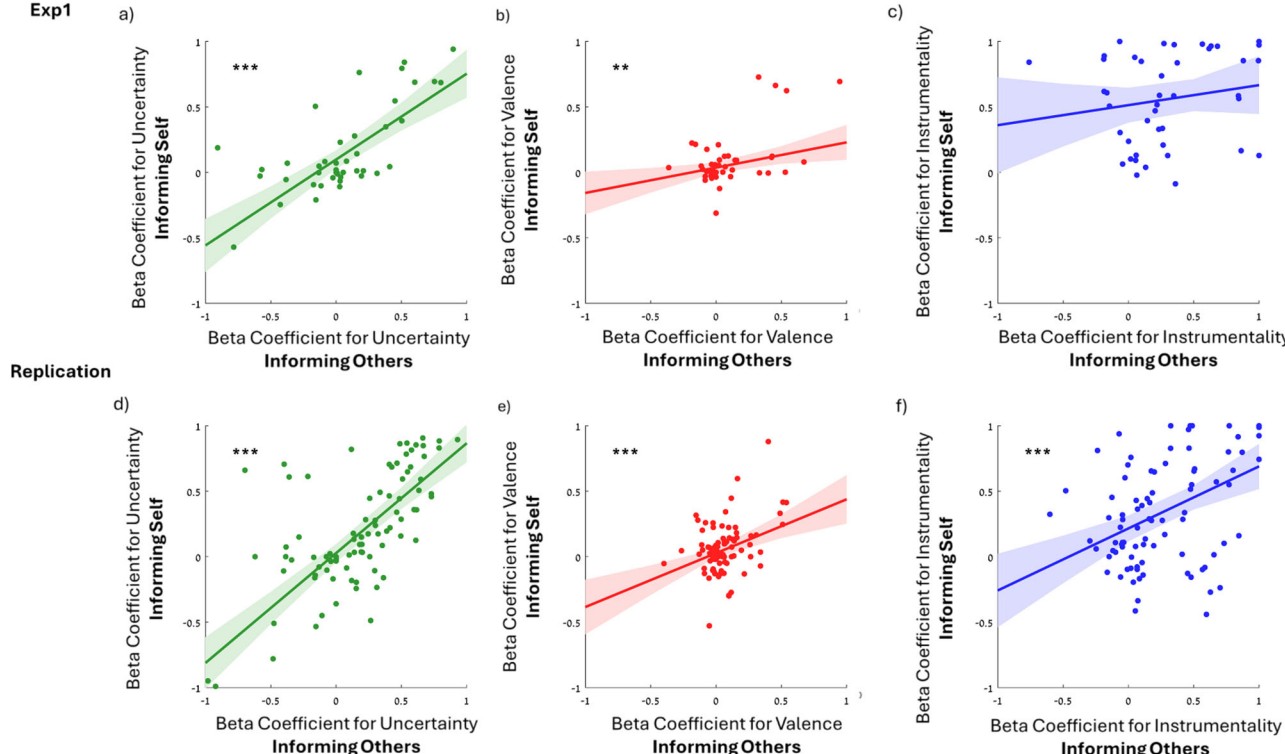

**Fig. 4 | Preferences are stable across information-seeking and information-sharing.** The robust correlations between the beta coefficient obtained when predicting information-seeking (x-axis) and information-sharing (y-axis) from **a**, **d** uncertainty, **b**, **e** valence, and **c**, **f** instrumentality in Exp. 1 (N = 43, top row) and the Replication (N = 90, bottom row). **a**, **d** Participants who preferred to seek information under high uncertainty also preferred to share information when the receiver was under high uncertainty. **b**, **e** Participants who preferred to seek positive information also preferred to share positive information. **c**, **f** Participants who preferred to seek useful information also preferred to share useful information (note, the effect is only statistically significant in the replication, where the N is larger). Lines represent robust regression lines. Smooth areas represent the confidence interval. ** = $p < 0.01$, *** = $p < 0.001$.

We then checked whether people who belong to one cluster when seeking information were more likely to belong to the analogs cluster when sharing it. Results showed that more than 62.8% (in Exp1) and 61.1% (in the replication study) of participants who were classified in one cluster when seeking information were classified in the analogous cluster when sharing it. A Binomial test revealed that this is significantly greater (Exp1: $p < 0.001$, Rep: $p < 0.001$) than chance (chance is 33%).

## Discussion

Our results suggest that people use their own information preferences to solve complex information-sharing decisions. In particular, when deciding whether to inform others, participants assigned similar relative weights to the usefulness of information, its valence, and the level of uncertainty as they do when deciding whether to seek information for themselves. These results suggest that people likely put themselves in the receiver's position to determine whether to inform others. Specifically, we found that participants shared information more when (i) the recipient could use it to gain rewards and avoid losses (i.e., when it had instrumental utility), (ii) when it was good news for the recipient rather than bad news and (iii) when the recipient was under high uncertainty. Sharers used the same rules to decide when to share information as they did to decide when to seek information. Importantly, the results suggest that they implement those rules from the point of view of the recipient, not their own. They seemed to consider the *recipients'* level of uncertainty, whether the *recipient* can use the information and how the information would make the *recipient* feel. They clearly did not consider their own point of view when sharing, as they were always completely certain of the content of the information and could not use information in any way to better their material outcome. In fact, that is a fundamental

distinction between seeking and sharing information - the former involves uncertainty about the information's content, while the latter assumes full knowledge. Importantly, despite this difference, information sharers can still adopt the recipient's perspective and potentially anticipate their curiosity when deciding what to share.

One may wonder what motivated participants to share information, as there was no monetary incentive in the information-sharing task. Analysis of participants' explicit written responses when asked, "How did you decide whether to open the envelope for the other participants?" suggests that participants were motivated by the other participant's welfare. Many explicitly referenced the other participant (using words such as 'they', 'them', 'others', and 'the participant') and considered the material and emotional impact of the shared information on the other person (using words like 'disappointment' or 'good news') and expressed a desire to help ("to help others out without making them feel bad"). This is consistent with many studies showing that people will help others, including strangers, even when there is no clear and immediate benefit to the self[39].

Previous studies on information-*seeking* indicate large individual differences in what people want to know[3,7,40]. We have previously shown that these differences can be accounted for by the different weights people assign to different motives for seeking information[3,7]. People tend to overweight one motive over the rest. Here, we replicated this result for information-seeking and, more importantly, showed similar individual differences in sharing preferences. In particular, a cluster analysis revealed that participants could be classified into three groups—one group cared mostly about instrumentality when deciding whether to share information, another mostly about the receiver's uncertainty, and a third preferred to share information that was positive for which the receiver was relatively certain about. The different

weights people assign to these factors may help explain why different people will make vastly different decisions on whether to inform others.

We find that such individual differences are consistent across information-sharing and information-seeking decisions. In particular, the weight assigned to each factor when seeking information was correlated to that assigned to each factor when sharing information. That is, the more people care about valence, instrumental value, and uncertainty when seeking information, the more they care about these factors when informing others. We replicate all our findings in a pre-registered study using an independent sample. These results suggest that information-seeking and sharing may rely on similar cognitive and neural mechanisms.

## Limitations
While the three-factor theory of information seeking[1] has already been demonstrated across different tasks and domains[7,41,42] (including financial, social, and health decisions), this study aims to test the three-factor theory for information-sharing. If indeed seeking and sharing decisions are closely related, we would expect our results to generalize to other domains and to naturalistic tasks. These predictions will need to be tested in future studies. For example, it would be interesting to examine information-sharing in situations where subjects are familiar with the recipients and/or on social media platforms. Further, it is interesting to consider if the three-factor theory could explain the finding that moral outrage leads to more information sharing on Twitter[43]. It is possible that users share content that elicits moral outrage because they believe that content is useful in guiding people's actions (e.g., has high instrumental utility, for instance by encouraging voting) even if it elicits negative emotions. Alternatively, users may believe the receiver will assess this type of information positively. For example, a statement like 'Trump is a liar' may well elicit a positive response from receivers who are not Trump supporters. That is to say, content that is categorized as eliciting 'moral outrage' may nonetheless be viewed as 'good news'. These different predictions can be tested by probing user's beliefs.

## Conclusions
Thanks to advances in technology, massive amounts of information are now easily accessible. This includes personalized information that can provide clues about a person's future health and finances. It is important to understand how people decide when to share such information, especially when they are not intimately familiar with the recipients or their preferences (a common scenario online). Here, we show that people consider the valence of information, its instrumental utility, and the receiver's uncertainty. People combine these estimates into a calculation of the value of information that can guide information-sharing choices. Our findings can help predict which information will be shared and help in framing critical information (such as health and safety) to increase the likelihood that it will be shared by others.

## Data availability
Anonymized data are available at a dedicated Github repository.

## Code availability
Code is available at a dedicated Github repository.

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

## Acknowledgements

We thank Bastien Blain, Irene Cogliati Dezza, Laura K. Globig, Christopher. A. Kelly, Sarah Zheng, and Sharon Machado Sanchez for comments on previous versions of the paper. The work was funded by a Wellcome Trust Senior Research Fellowship (214268/Z/18/Z) to T.S. The funders had no role in study design, data collection and analysis, the decision to publish, or the preparation of the paper.

## Author contributions

Valentina Vellani: Conceptualization, Methodology, Formal Analysis, Investigation, Writing—Review & Editing, Visualization; Moshe Glickman: Formal Analysis, Writing—Review & Editing, Visualization; Tali Sharot: Conceptualization, Methodology, Writing—Review & Editing, Supervision.

## Competing interests

The authors declare no competing interests
