## [Peer Review File · Communications Psychology]

7th Aug 23

Dear Dr Vellani,

Thank you for your patience during the peer-review process. Your manuscript titled "How people decide when to inform others" has now been seen by 3 reviewers, and I include their comments at the end of this message. You will see that they find your work of some potential interest. However, they have raised quite substantial concerns that must be addressed. In light of these comments, we cannot accept the manuscript for publication, but would be interested in considering a revised version that fully addresses these serious concerns.

We hope you will find the Reviewers' comments useful as you decide how to proceed. Should additional work allow you to address these criticisms, we would be happy to look at a substantially revised manuscript. If you choose to take up this option, please highlight all changes in the manuscript text file, and provide a detailed point-by-point reply to the reviewers.

Editorially, we consider it important that the revision addresses the reviewers' concerns regarding the selection of statistical analyses, as well as the incorporation of additional analyses as suggested by the reviewers.

Especially given the significant concerns voiced by Reviewer #1 regarding the lack of clarity regarding the participants' goals in the information-sharing task, we ask you to ensure that the presentation of the findings stays close to the data and does not go beyond what can be demonstrated in the current studies. Please provide additional details on how the studies were presented to the participants. It is critical that you provide an appropriate limitations section that addresses the caveats and limitations of your work and the scope of the contributions of the current findings.

For manuscripts that report null results, we require the following:

- Evidence that the study is sufficiently powered to detect the smallest theoretically or pragmatically meaningful effect

- Bayes Factors or equivalence tests to interpret the null results

- Appropriate language to describe the results. (There is no statistical test that can demonstrate absence of an effect. Statements such as 'There is no difference between x and y.' or 'X does not affect Y.' must be revised to read 'We found [no/little] credible evidence of a difference between x and y.' or 'We found [no/little] credible evidence that X affects Y.')

Please use the following link to submit your revised manuscript, point-by-point response to the referees' comments (which should be in a separate document to any cover letter) and the completed checklist:

[link redacted]

Please do not hesitate to contact me if you have any questions or would like to discuss these revisions further. We look forward to seeing the revised manuscript and thank you for the opportunity to review your work.

Best regards,

Patricia Lockwood

Patricia Lockwood, PhD

Editorial Board Member

Communications Psychology

orcid.org/0000-0001-7195-9559

EDITORIAL POLICIES AND FORMATTING

Editorial Policy: Policy requirements (Download the link to your computer as a PDF.)

Furthermore, please align your manuscript with our format requirements, which are summarized on the following checklist:

Communications Psychology formatting checklist

and also in our style and formatting guide Communications Psychology formatting guide .

* **CODE AVAILABILITY:** All Communications Psychology manuscripts must include a section titled "Code Availability" at the end of the methods section. In the event of publication, we require that the custom analysis code supporting your conclusions is made available in a publicly accessible repository; at publication, we ask you to choose a repository that provides a DOI for the code; the link to the repository and the DOI will need to be included in the Code Availability statement. Publication as Supplementary Information will not suffice. We ask you to prepare code at this stage, to avoid delays later on in the process.

* **DATA AVAILABILITY:**

All Communications Psychology manuscripts must include a section titled "Data Availability" at the end of the Methods section or main text (if no Methods). More information on this policy, is available at <http://www.nature.com/authors/policies/data/data-availability-statements-data-citations.pdf>.

At a minimum the Data availability statement must explain how the data can be obtained and whether there are any restrictions on data sharing. Communications Psychology strongly endorses open sharing of data. If you do make your data openly available, please include in the statement:

We recommend submitting the data to discipline-specific, community-recognized repositories, where possible and a list of recommended repositories is provided at <http://www.nature.com/sdata/policies/repositories>.

If a community resource is unavailable, data can be submitted to generalist repositories such as figshare or Dryad Digital Repository. Please provide a unique identifier for the data (for example a DOI or a permanent URL) in the data availability statement, if possible. If the repository does not provide identifiers, we encourage authors to supply the search terms that will return the data. For data that have been obtained from publicly available sources, please provide a URL and the specific data product name in the data availability statement. Data with a DOI should be further cited in the methods reference section.

REVIEWERS' EXPERTISE:

Reviewer #1 Decision-making, learning, linear mixed effects models

Reviewer #2 Decision-making, social cognition, linear mixed effects models

Reviewer #3 Information seeking, decision-making

REVIEWERS' COMMENTS:

Reviewer #1 (Remarks to the Author):

In this manuscript, Vellani and Sharot investigate how people decide when to inform others. They use 2 symmetric experimental designs, based on a financial market task, aimed at exploring how individuals seek information for themselves and/or share information to others. They manipulate 3 factors hypothesized to affect levels of information seeking/sharing: the usefulness of information in directing future action, its valence and the receiver's uncertainty level. They report that participants are indeed sensitive to these 3 factors, similarly in the information seeking and information sharing task.

Overall, this work seems interesting but also somewhat limited. On the one hand, the topic is of interest, the task well designed and the pre-registered replication strengthen the conclusions. On the other hand, the lack of control on participants' actual goal, the superfluous aspect of some analyses and the shortcomings of others, limit the scope of the contribution.

I list below my main concerns, and hope that the authors will find those useful to improve their manuscript.

Main issues.

The first, major line of issues is conceptual and directly impacts the interpretation of the findings. In short, it seems that participants have no explicit goal in the information-sharing task. Although the abstract and introduction evoke "solv[ing] complex information-sharing problems", there does not seem to be a particular problem to solve: participants' payoff does not seem to depend on the

payoff of the information recipient, interactions do not seem reciprocal, etc. so there does not seem to be any real consequences of information sharing (or absence thereof).

1. From there, I find it hard to interpret what participants are doing – and I don't find really surprising that, in the absence of any goal or instruction, they would follow a strategy to solve the closest and most intuitive task: the information seeking task (i.e. behave as if they would be the recipient). Is there any way to know what participants are trying to achieve in the information sharing task?
2. Are participants really playing with real recipient? If not, what is the cover story, and is it credible? Was this checked? If not, it becomes even harder to interpret the finding with the angle of altruistic motivations and actual, genuine preferences towards information sharing.
3. Related to this first point, no information is given in the methods about the incentivization/payoff scheme in the information sharing task.
4. Also, I don't understand the information seeking incentivization/payoff scheme written lines 498-499 "Recipients were informed that they would start the task with 250K bonus points which were worth between £1 and £5 together." So, what is the value of Experimental units? Does that vary between individuals? Or was it left ambiguous (in which case, participants might form different beliefs about it). This could be an issue, given that several analyses explore inter-individual differences...

The second, line of issues is methodological. In short, it seems that some analyses are incomplete or somewhat superfluous.

5. Why are there no interaction between factors in the linear regressions? Currently, the model comparison seems redundant with the test on the regression coefficients, and it could be of better use to guide the inclusion/exclusion of interaction terms, which are a bit more exploratory. In general, the model comparison is also not very principled: the use of AIC and fixed-effect approach (i.e. conflating/summing within and between subject dimensions) is known to inflate type 2 errors (i.e. both lead to the selection over-complex models).
6. What is the actual predictive power of the model? (pseudo)-R²? To improve the interpretation of the different effects and of the model fit, would not it be more informative to picture the observed vs predicted levels of information sharing/seeking across the dimensions of the 3 experimental factors rather than having a Figure 3 that pictures the regression parameters which are already reported in Table 1 and Table 2 ?
7. The correspondence between the fully-specified mixed models and the summary-statistic approach based on individual fits also seem unnecessary in the Main Text (it should be a given) and could be send to a supplementary document.
8. I'm a bit puzzled by the clustering approach, which seem a bit inconclusive, IMHO. By imposing K=3, the analysis does not rigorously test the presence of categorical clusters (one should make a model comparison between K = 1, 2, 3, 4, ...). The fact that the three "clusters" seem to simply leverage the 3 experimental dimensions is also not a good sign, regarding the inferential value of the exercise.

9. The clustering analysis also looks disconnected from the rest of the paper, which claim that the drivers of behavior are similar in information-seeking and information sharing task: do participants who performed both tasks get assigned to a similar cluster in both tasks? If yes, what is the value of the subsequent between-task regression? If not, why would the 2 analyses give different conclusions about the robustness of the information preference type?

Minor:

10. I find the description of what "Algorithm accuracy" is and how it was implemented very unclear/confusing. For the prediction of a continuous variables, what does "correct X% of the time" means? Does it mean that it gives the true estimate x% of the time, and makes predictions uniformly distributed over the support (-400 to -500 or +400 to +500) the rest of the time (100-x%)?

Reviewer #2 (Remarks to the Author):

This is an interesting and timely study examining to what extent uncertainty, valence and instrumentality drive information seeking and sharing. The main conclusion is that the individual preferences in information seeking also drive information sharing. The findings were replicated in an independent study, further strengthening the conclusions.

While I enjoyed reading the manuscript and I am overall positive, I do think deeper insight should be gained from the data by including additional analyses and plots.

I would expect an interaction between instrumentality and valence. For example, do people especially seek out positive information/avoid negative information when they cannot change the outcome anyway? This can be easily tested by including interaction terms in the analysis.

The term "valence" is often only used to indicate whether an outcome was positive or negative. Did the authors mean magnitude * valence in the analyses? If magnitude wasn't part of the analyses I would like to see it included as well because that may well explain part of the variance. Adding these as separate terms in the regression would tease apart those effects.

Figure 2. Based on AIC/BIC values alone we don't know whether the best model was significantly better than the others. One option is to calculate the 95%CI of the BIC difference using bootstrapping.

Figure 3 is not very informative to evaluate the model predictions as it mostly shows that there were many individual differences in the beta weights. Instead, it may be more insightful to plot the eagerness to see/share the information as a function of the beta's.

Figure 5. This is a key figure but some of the relationships look a little exaggerated by influential datapoints (e.g., fig5 b, c and f). It would be better to test the robustness against influential datapoints using robust linear mixed effects models and robust regression methods to draw the fit line.

Interestingly, it seems from figure 5 that the beta weight for instrumentality for information seeking is least correlated to the beta weight for information sharing compared to the self-other correlations for uncertainty and valence beta weights. Can the authors test if this is statistically true and if so, discuss what might drive this finding?

I would like to see some discussion on how the findings fit with the literature on information sharing in real life. For example, some studies suggest that moral outrage drives information sharing on twitter, while this study mostly finds that positivity drives information seeking/sharing. I would like to see some discussion as to where this discrepancy between the results and the "real-life" findings come from.

Minor points

Please clarify the task design choices. For example, if participants indicated that they wanted to open the envelope, they were "likely" to gain access to the requested information. Why was this not deterministic and with what probability did they get to see the information? How would the results change if this would've been deterministic?

What was the cost of information seeking/sharing? Why would participants not always look/share?

When discussing the results, I got the impression that the dependent variable was binary (sharing yes/no and seeking yes/no). But from the task description it seems that eagerness to share/seek

was measured on a scale from 1-6, so I assume it was not binary. Please explain what the dependent variable was and change the text accordingly.

Was any questionnaire data collected on individual differences that could be related to information seeking/sharing preferences? For example, positivity/negativity bias or anhedonia?

Reviewer #3 (Remarks to the Author):

This paper draws a connection between information seeking and information sharing, finding in an online experiment (and a replication thereof) that the propensity to share information is increasing in (prediction) uncertainty, information instrumentality, and positive valence. A key result is that, among those participants who performed both an information seeking and information sharing task (deciding to open or reveal information about their “stock” performance in the experimental “stock market”), there exists a strong correlation between the weight assigned to each factor for both task types (i.e., the relative influence of each factor is stable across information-seeking and information-sharing decisions). This work presents an interesting extension of and contribution to the existing literature. There are, however, several issues that should be addressed, mostly in terms of acknowledging the limitations and making the work more accessible.

1. With respect to former point, there is no discussion of the work’s limitations. I highlight a few notable ones below, but careful consideration should be given to the possibility of others as well.

a. External validity and generalizability: As with any experimental setting, the context is sterile and neglects many important features of real-world judgment, decision-making, and behavior. For instance, nothing is known to the focal participant of the other “player” so participants are limited to the three factors of interest (i.e., uncertainty, instrumentality, affective impact) and their own information-seeking preferences to make their information-sharing decision. (The authors somewhat acknowledge this in the line beginning with “Future studies should test the generatability [sic] of these findings...” but it seems a more critical issue and potential alternative explanation for their results than to hand-wave it off as beyond the scope of this research.) Additionally, in the case of stocks, the factors are unambiguous. Would the information value calculation hold in other contexts, for example, socially relevant information?

b. Fatigue/heuristic responding: It seems participants are asked to make a substantial number of decisions across multiple blocks of 40 trials each. Fatigue is always a concern in such situations. It is also possible that the 55 and 108 participants (Exp 1 and rep) who completed both tasks established a pattern of responding in the first block that they simply carried over to the second, more for ease of responding than due to a link between their information seeking vs. sharing tendencies.

c. Ambiguity: Likely in the interest of avoiding deception, the authors introduce some language that is a bit ambiguous – notably, when they talk about the “recipient” who “may be playing a similar task tomorrow.” This likely introduces some ambiguity and may lead participants to doubt the credibility that there is in fact a real “player” with whom they are sharing the information. This possibility again threatens the authors’ result that individuals project their own information seeking preferences onto others, i.e., if they don’t really think there is an actual “other.”

d. Task differences: It seems that the two experimental tasks are fundamentally different: “Opening the envelop” in the information-seeking task involves an element of curiosity absent from the information-sharing task, where the participant always has complete information. This seems worth noting in the paper.

The above issues do not render the work unworthy of publication or, in my opinion, demand additional data, but they do necessitate discussion in the current paper.

2. The paper can be made far more accessible with a re-organization and careful attention to gaps in the reader’s knowledge – a tricky endeavor when you’re so close to the research design but very necessary.

a. The paper would benefit immensely from an upfront roadmap of where the authors are heading, explicit discussion of the specific hypotheses or research goals, and what specific results would be interpreted as support for their predictions. In particular, the finding that individuals prefer to share positive, instrumental, uncertain information is not especially surprising, but the relative weights assigned to each and the relationship between information-seeking vs. -sharing is interesting. It would be good if the authors discussed up front how, specifically, they planned to tease this latter point out given their data.

b. The section currently labeled “Results” seems actually to be the Methods/Procedure section. Its location is good, and indeed I would highly recommend moving the later Methods/Procedures up, as many of those details are important for the reader to appreciate the methodology (e.g., who are the participants, is decision making incentive compatible, are blocks counterbalanced, etc.). In particular, participants are told “If they indicated they wanted to open the envelope they were then LIKELY to be shown the current value of their stocks (information).” Upon reading this, one immediately wonders what this means in terms of probability and how this probability relates to participant ratings. Providing all relevant information up-front would be very helpful.

(Note: I think the caption for Figure 1 is actually a clearer, more succinct way of articulating what is presented in the five preceding paragraphs. If the authors are concerned about space with relocating the detailed information from the current Methods/Procedure sections to the body of the paper, this is one way of consolidating.)

c. The “Results” section seems actually to begin with the sentence “Participants consider the impact of information on affect, action and uncertainty when deciding whether to inform others” and should thus be labeled as such.

d. The discussion and interpretation of the (AIC, regression, etc.) results and figures is quite limited and the paper would benefit from a more meaningful discussion thereof.

e. Somewhat relatedly, presenting the raw number of participants from each experiment falling into a given cluster seems less effective than presenting those values as percentages, which would tell the reading something about which cluster is more common.

f. The paper has a fair number of typos that should, of course, be corrected (e.g., see [sic] above).

3. Finally, a few points of clarification:

a. Do analyses include controls for things like block order and variable value for a given trial?

b. I'm confused by the sentence beginning with “Examining the weight each participant assigned to instrumentality in both tasks revealed that participants who assigned greater weight to a particular factor when seeking information themselves also assigned a greater weight to that factor when deciding whether to share information...” Were some participants not assigned to an instrumentality block? If not, how can you have a coefficient for Instrumentality for those participants? Sorry if I missed something here...

We would like to thank the reviewers for a thorough and thoughtful commentary on our work. We were encouraged by the positive assessment of the reviewers, who noted that “*the topic is of interest, the task well designed and the pre-registered replication strengthen the conclusions*” [Reviewer 1], that “*this work presents an interesting extension of and contribution to the existing literature*” [Reviewer 3], and that “*This is an interesting and timely study examining to what extent uncertainty, valence and instrumentality drive information seeking and sharing*” [Reviewer 2]. The reviewers also provided helpful suggestions and comments, all of which we were able to address in the revised manuscript. First, we added a new analysis that elucidates participants’ goals in the information-sharing task. Second, we examined two and three-way interactions between instrumentality, uncertainty and valence in the information seeking and sharing tasks. Third, we carefully revised text and figures for improved clarity. Additionally, we addressed all other reviewer’s comments and suggestions. To ensure we address all the Reviewers’ comments, and for ease of reference, we include the reviews below (in bold) followed by our response to each concern.

Reviewer #1:

In this manuscript, Vellani and Sharot investigate how people decide when to inform others. They use 2 symmetric experimental designs, based on a financial market task, aimed at exploring how individuals seek information for themselves and/or share information to others. They manipulate 3 factors hypothesized to affect levels of information seeking/sharing: the usefulness of information in directing future action, its valence and the receiver’s uncertainty level. They report that participants are indeed sensitive to these 3 factors, similarly in the information seeking and information sharing task.

Overall, this work seems interesting but also somewhat limited. On the one hand, the topic is of interest, the task well designed and the pre-registered replication strengthen the conclusions. On the other hand, the lack of control on participants’ actual goal, the superfluous aspect of some analyses and the shortcomings of others, limit the scope of the contribution. I list below my main concerns, and hope that the authors will find those useful to improve their manuscript.

- We thank the reviewer for their useful and thoughtful feedback, which helped to improve the manuscript. As detailed below, we provide new data and analysis clarifying the goals of the participants in the information sharing task. We also refined both the clustering and regression analyses. We believe that addressing these elements has further strengthened our study.

Main issues.

The first, major line of issues is conceptual and directly impacts the interpretation of the findings. In short, it seems that participants have no explicit goal in the information-sharing task. Although the abstract and introduction evoke “solv[ing] complex information-sharing problems”, there does not seem to be a particular problem to solve: participants’ payoff does not seem to depend on the payoff of the information recipient, interactions do not seem reciprocal, etc. so there does not seem to be any real consequences of information sharing (or absence thereof).

From there, I find it hard to interpret what participants are doing – and I don’t find really surprising that, in the absence of any goal or instruction, they would follow a strategy to solve the closest and most intuitive task: the information seeking task (i.e. behave as if they would be the recipient). Is there any way to know what participants are trying to achieve in the information sharing task?

- We thank the reviewer for prompting us to address this point. Indeed, we did collect data that sheds light on participants’ goal in the information-sharing task. Specifically, at the end of Exp1, 69 participants were asked to describe how they approached the information-sharing task (“How did you decide whether to open the envelope for the other participants?”). In response, the majority of participants explicitly referenced the other participant (using words such as ‘they’, ‘them’, ‘others’ or

‘the participant’) and explicitly mentioned the instrumental and non-instrumental impact of the shared information on the other participant (using the words like ‘surprise’, ‘disappointment’, ‘good news’, ‘bad news’, value of ‘knowing’). A desire to help was also expressed frequently by the participants.

Below a few representative responses:

- “In the first task my decision was based off whether *the participants would be disappointed or pleased*. The second task was whether *they should risk investing their money*.”
- “Good news vs bad news, reinforce if *they were doing well* and encourage where performing above expectations.”
- “I wanted to give the participants the most info I could.”
- “I’m a nice person if it made a difference, I wanted them to see it, the first part it didn’t matter so I didn’t care.”
- “Mostly if the participant would have the chance to move from a negative scenario to a positive which was possible *only with my help*.”
- “To *help* others out without making them feel bad”

These responses strongly suggest that participants aimed to help others by providing information they believed would be beneficial to them (in terms of money, emotion, etc.). This is consistent with vast research demonstrating that people often help others even if there is no material benefit to them, as they derive positive feelings from doing so (Krebs, 1975; Warneken, F., & Tomasello, M. 2009).

- We quantified the above interpretation by having two naïve observers independently rate all responses to indicate whether the responder:

(1) believed they were interacting with another participant. (Options: (a) the responder believes they are interacting with another participant (56.5%); (b) the responder does not believe they are interacting with another participant (0%); (c) no indication either way (43.5%).)

(2) is trying to help the other participant. (Options: (a) the responder is trying to help the other participant (51.5%); (b) the responder is not trying to help the other participant (2.9%); (c) no indication either way (45.6%).)

The raters showed strong agreement between them (average absolute ICCs was 0.97 for question 1 and 0.96 for question 2). Analysis of the ratings of the first question revealed that 56.5% of responders explicitly indicated that they believed they were interacting with another participant; 0% of responders explicitly indicated that they did not believe they were interacting with another participant; 43.5% of responses were not informative either way. Analysis of the ratings of the second question revealed 51.5% of responses explicitly demonstrated a desire to help; 2.9% of responders explicitly indicated no desire to help; 45.6% of responses were not informative either way.

- In sum, the new findings support the assumption that a large fraction of participants’ goal was to help others by sharing information they considered beneficial. We have now included these results in the revised manuscript (pg 9).

Are participants really playing with real recipient? If not, what is the cover story, and is it credible? Was this checked? If not, it becomes even harder to interpret the finding with the angle of altruistic motivations and actual, genuine preferences towards information sharing.

- The participants did not play with a real recipient, but were led to believe they were doing so. To assess participants’ belief in the cover story, we asked 69 participants how they approached the information-sharing task (“How did you decide whether to open the envelope for the other participants?”) and asked two naïve raters to independently rate all responses to indicate whether the responder:

(1) believed they were interacting with another participant. (Options: (a) the responder believes they are interacting with another participant; (b) the responder does not believe they are interacting with another participant; (b) no indication either way).

Analysis of the ratings of the first question revealed that 56.5% of responders explicitly indicated that they believed they were interacting with another participant; 0% of responders explicitly indicated that they did not believe they were interacting with another participant; 43.5% of responses were not informative either way (pg 9).

Related to this first point, no information is given in the methods about the incentivization/payoff scheme in the information sharing task.

- In the information sharing task there was no monetary incentive to share information with the other players. This is now specified in the manuscript (pg. 5-6).

Also, I don't understand the information seeking incentivization/payoff scheme written lines 498-499 "Recipients were informed that they would start the task with 250K bonus points which were worth between £1 and £5 together." So, what is the value of Experimental units? Does that vary between individuals? Or was it left ambiguous (in which case, participants might form different beliefs about it). This could be an issue, given that several analyses explore inter-individual differences...

- All participants were informed that they will receive 250K bonus points which is worth between \$1 and \$5 (exact value was not provided, this is clarified now in pg. 5-6). This was done to avoid participants trying to calculate the value of information with maths. This is the same strategy used successfully in similar information-seeking studies (see Cogliati Dezza et al., 2022, Globig et al, 2021) and is selected because it better mirrors real-world decision-making scenarios where the cost of information is often difficult to quantify precisely (pg. 5).
- There are two analyses investigating individual differences. The first, investigates the correlation between seeking and sharing. Individual differences regarding monetary expectation (i.e. whether a unit is equal to \$1 or \$5) should be the same for info-seeking and info-sharing as the instructions are worded in a way that elicits no reason to conclude the sum alters across tasks. As a consequence, different monetary expectations likely account for the correlation between info-sharing and seeking preferences. The second is the cluster analyses. Here, participants who have higher expectations regarding monetary rewards could put more weight on instrumentality and thus would be more likely to be in the 'instrumentally cluster'. Indeed, this could be an explanation for their preferences, not just here, but also in real-life where the magnitude of instrumentality (monetary or otherwise) is often unknown and reward expectations can alter information-seeking preferences. In fact, we have made this exact prediction elsewhere (Sharot & Sunstein, 2020).

The second, line of issues is methodological. In short, it seems that some analyses are incomplete or somewhat superfluous.

- We thank the reviewer for the below comments, addressing them helped us to improve the analyses.

Why are there no interaction between factors in the linear regressions?

- Following the reviewer's suggestion, we have now added interactions to the linear regressions. All main effects remain significant. We also find a weak interaction between instrumentality and valence in the information sharing in Exp.1 & Replication. The effect size of this interaction compared to the main effects is small and is not significantly different from the non-significant interaction for information-seeking (Exp. 1: $t(221) = 1.2$, $p = 0.23$ & Replication: $t(200) = 1.73$, $p = 0.08$). We now report the interactions in the supplementary material (pg. 4-5 of supplementary materials).

Currently, the model comparison seems redundant with the test on the regression coefficients, and it could be of better use to guide the inclusion/exclusion of interaction terms, which are a bit more exploratory.

- This is a good point. Following the reviewer's suggestion, we now only report results from the subject-by-subject linear regressions. For transparency, the mixed-model results are reported in the Supplementary Materials (pg. 1 of supplementary materials).

In general, the model comparison is also not very principled: the use of AIC and fixed-effect approach (i.e. conflating/summing within and between subject dimensions) is known to inflate type 2 errors (i.e. both lead to the selection over-complex models).

- In this study we used Linear Mixed Effect models that includes both fixed and random effects, which do not inflate type 2 errors.
- We now report BIC instead of AIC in the Supplementary Results (pg. 3 of supplementary materials)..

What is the actual predictive power of the model? (pseudo)-R2?

- The median R^2 scores for information-seeking: Exp1: median $R^2 = 0.51$, $SE = 0.03$, Replication: median $R^2 = 0.46$, $SE = 0.03$; and for sharing: Exp1: median $R^2 = 0.37$, $SE = 0.02$, Replication: median $R^2 = 0.40$, $SE = 0.03$ (pg. 9).

To improve the interpretation of the different effects and of the model fit, would not it be more informative to picture the observed vs predicted levels of information sharing/seeking across the dimensions of the 3 experimental factors rather than having a Figure 3 that pictures the regression parameters which are already reported in Table 1 and Table 2?

- We thank the reviewer for this suggestion, which prompted us to replace Fig.3 with scatterplots illustrating the relationship between the z-scored observed and predicted information seeking/sharing in Experiment 1 and Rep now reported in the Supplementary (FigS1, pg. 2 of supplementary materials). The scatterplots are presented separately for the different motives: Uncertainty, Valence and Instrumentality.

The correspondence between the fully-specified mixed models and the summary-statistic approach based on individual fits also seem unnecessary in the Main Text (it should be a given) and could be send to a supplementary document.

- This has now been moved to Supplementary Materials (pg 1 of supplementary materials).

I'm a bit puzzled by the clustering approach, which seem a bit inconclusive, IMHO. By imposing K=3, the analysis does not rigorously test the presence of categorical clusters (one should make a model comparison between K = 1, 2, 3, 4, ...). The fact that the three "clusters" seem to simply leverage the 3 experimental dimensions is also not a good sign, regarding the inferential value of the exercise.

- We appreciate the reviewer's suggestion to refine our clustering approach. To address this, we implemented the K-means clustering algorithm, while varying K between 2 to 7 (K=1 was omitted, as the algorithm requires at least 2 clusters). To evaluate the quality of clustering for each K, we calculated the Calinski-Harabasz index (Caliński, T., & Harabasz, 1974). **Table 2** displays these indices for each dataset as a function of K. Across datasets K=3 consistently yielded the optimal clustering solution. We report these results on page 16.
- Importantly, we could not predict the optimal K nor the distribution of betas within each cluster a priori. Our findings reveal two clusters characterized by a single dominant motive (uncertainty and instrumentality, see Fig. 2), and a third, more complex cluster, which is characterized by a positive weight for valence, but negative weight for uncertainty (especially when information is shared). This

highlights that information-seeking/sharing preferences can be driven by individual factors or a mixture, offering a more nuanced view than focusing solely on dominant motivations.

The clustering analysis also looks disconnected from the rest of the paper, which claim that the drivers of behavior are similar in information-seeking and information sharing task: do participants who performed both tasks get assigned to a similar cluster in both tasks? If yes, what is the value of the subsequent between-task regression? If not, why would the 2 analyses give different conclusions about the robustness of the information preference type?

- Indeed, 62.8% (in Exp1) and 61.1% (in the Replication study) of participants retained their cluster assignment across tasks. A Binomial test revealed that this is significantly greater than chance (Exp1: $p < 0.001$, Rep: $p < 0.001$, (chance is 33.3%) (pg 14). We believe that having two very different analysis (clustering and cross subjects' correlation) that indicate the same underlying pattern is a strength of the paper and we opt to keep them both.

Minor:

I find the description of what "Algorithm accuracy" is and how it was implemented very unclear/confusing. For the prediction of a continuous variables, what does "correct X% of the time" means? Does it mean that it gives the true estimate x% of the time, and makes predictions uniformly distributed over the support (-400 to -500 or +400 to +500) the rest of the time (100-x%)?

- Exactly, it gives the true estimate x% of the time, and makes predictions uniformly distributed over the rest (-400 to -500 or +400 to +500) the rest of the time (100-x%). We now clarify this in the manuscript (pg. 5).

Reviewer #2 (Remarks to the Author):

This is an interesting and timely study examining to what extent uncertainty, valence and instrumentality drive information seeking and sharing. The main conclusion is that the individual preferences in information seeking also drive information sharing. The findings were replicated in an independent study, further strengthening the conclusions. While I enjoyed reading the manuscript and I am overall positive, I do think deeper insight should be gained from the data by including additional analyses and plots.

- We thank the reviewer for this positive assessment.

I would expect an interaction between instrumentality and valence. For example, do people especially seek out positive information/avoid negative information when they cannot change the outcome anyway? This can be easily tested by including interaction terms in the analysis.

- Following the reviewer's suggestion, we have now added interactions to the linear regressions. All main effects remain significant. We also find a weak interaction between instrumentality and valence in the information in Exp.1 & Replication. The effect size of this interaction compared to the main effect is small and the interaction is not significantly different from the non-significant interaction for information-seeking (Exp. 1: $p = 0.23$ & Replication: $p = 0.08$). We now report the interactions in the Supplementary Material (pg. 4-5 of supplementary materials).

The term "valence" is often only used to indicate whether an outcome was positive or negative. Did the authors mean magnitude * valence in the analyses? If magnitude wasn't part of the analyses I would like to see it included as well because that may well explain part of the variance. Adding these as separate terms in the regression would tease apart those effects.

Indeed, when we say ‘valence’ we mean magnitude * valence. We now clarify this in the revised manuscript (pg. 5).

Figure 2. Based on AIC/BIC values alone we don’t know whether the best model was significantly better than the others. One option is to calculate the 95%CI of the BIC difference using bootstrapping.

- We thank the reviewer for their comment, which prompted us to clarify the meaning of the BIC scores in our results. Differences in BIC scores between two models approximate Bayes factor, according to the formula (Shen & González, 2021; Wagenmakers, 2007):

$$BF = e^{-\frac{1}{2}(BIC_{model1} - BIC_{model2})}$$

A BIC difference greater than 10 indicates very strong evidence in favour of the model with the lower score (see Raftery, 1995, Lorah, J., & Womack, A., 2019). To make it easier to compare the differences between the different models and the best model in all of our experiments, we have added a new column to Table S2, which indicates the BIC difference from the best (full) model (pg. 3 in supplementary results).

Figure 3 is not very informative to evaluate the model predictions as it mostly shows that there were many individual differences in the beta weights. Instead, it may be more insightful to plot the eagerness to see/share the information as a function of the beta’s.

- We thank the reviewer for this suggestion, which prompted us to replace Fig.3 with scatterplots illustrating the relationship between the z-scored observed and predicted information seeking/sharing in Experiment 1 and Rep now reported in the Supplementary (Fig S1, pg. 2 in supplementary results). The scatterplots are presented separately for the different motives: Uncertainty, Valence and Instrumentality.

Figure 5. This is a key figure but some of the relationships look a little exaggerated by influential datapoints (e.g., fig5 b, c and f). It would be better to test the robustness against influential datapoints using robust linear mixed effects models and robust regression methods to draw the fit line.

- We thank the reviewer for this comment. As suggested, we repeated all the analyses in Fig.5 using Robust Regression (Huber) and we now show the fit line. The results remained essentially the same: Exp. 1: Uncertainty: $b = 0.66$, $t(41) = 7.38$, $p < 0.001$, Valence: $b = 0.19$, $t(41) = 2.72$, $p = 0.009$, Instrumentality: $b = 0.15$, $t(41) = 1.13$, $p = 0.26$, Replication: Uncertainty $b = 0.83$, $t(88) = 10.52$, $p < 0.001$, Valence: $b = 0.41$, $t(88) = 4.21$, $p < 0.001$, Instrumentality: $b = 0.47$, $t(88) = 4.47$, $p < 0.001$. The robust regression results are reported in the Results (pg. 13-14).

Interestingly, it seems from figure 5 that the beta weight for instrumentality for information seeking is least correlated to the beta weight for information sharing compared to the self-other correlations for uncertainty and valence beta weights. Can the authors test if this is statistically true and if so, discuss what might drive this finding?

- Following the reviewer's suggestion, we compared the correlations between the betas coefficients for seeking and sharing behaviours across variables. The results revealed that the correlation between uncertainty coefficients were greater than those of valence (Exp1: $z = 1.33$, trend $p = 0.09$, Rep: $z = 2.47$, $p = 0.006$) and instrumentality (Exp1: $z = 3.2$, $p < 0.001$, Rep: $z = 2.24$, $p = 0.01$). For Exp1 the correlation between valence coefficients was greater than that of instrumentality ($z = 1.87$, $p = 0.03$), while there was no difference between the two in the replication ($z = -0.23$, $p = 0.40$).

I would like to see some discussion on how the findings fit with the literature on information sharing in real life. For example, some studies suggest that moral outrage drives information sharing on

twitter, while this study mostly finds that positivity drives information seeking/sharing. I would like to see some discussion as to where this discrepancy between the results and the “real-life” findings come from.

- Our study suggest that information-sharing is driven by a complex interplay of perceived instrumental utility, uncertainty reduction and affective considerations. With regards to the valence of information, what is considered good news vs. bad news is context-dependent. Seemingly negative information (in terms of its emotional content) may be shared, if the sharer believes the receiver will assess the information in a positive way. For example, a statement like ‘Trump is a liar and should be jailed’ may elicit a positive response in receivers who are not Trump supporters. Therefore, content that may trigger ‘moral outrage’ may nonetheless be viewed positively by some recipients. Moreover, our study absolutely suggest that people will indeed share negative information if they believe it is instrumental and/or reduces uncertainty. We now clarify this in the manuscript (pg. 15-16).

Minor points

Please clarify the task design choices. For example, if participants indicated that they wanted to open the envelope, they were “likely” to gain access to the requested information. Why was this not deterministic

- This feature was selected as it was adapted from past information-seeking studies (Charpentier et al., 2018, Vellani et al., 2020). The probabilistic nature simply reflects real-life, where outcomes are rarely completely deterministic. We now clarify this in the manuscript (pg 5).

and with what probability did they get to see the information?

- If participants selected 0, 1, 2, 3, 4, 5, 6 then information was delivered with a probability of 5%, 20%, 35%, 50%, 65%, 80%, 95%, respectively (pg. 5).

How would the results change if this would’ve been deterministic?

- We speculate they would not alter. This speculation is based on the fact that for information-seeking studies tasks using deterministic outcomes (Kelly and Sharot 2021, Golman et al., 2017; Hertwig & Engel, 2016; Karlsson et al., 2009), show similar results to those using probabilistic outcomes (Charpentier et al., 2018, Vellani et al., 2020).

What was the cost of information seeking/sharing? Why would participants not always look/share?

- In this study, there was no cost associated with sharing or seeking decisions. The results show that participants are less likely to share/seek when the information is negative and thus elicit negative emotions and also when there is no instrumental utility to information and when information is practically already known (certainty is high) (pg. 9 - see also Charpentier et al., 2018, Vellani et al., 2020, Cogliati Dezza et al., 2022). The latter two maybe in order to avoid information overload.

When discussing the results, I got the impression that the dependent variable was binary (sharing yes/no and seeking yes/no). But from the task description it seems that eagerness to share/seek was measured on a scale from 1-6, so I assume it was not binary. Please explain what the dependent variable was and change the text accordingly.

- Recipients indicated whether they wanted to open an envelope using a seven points Likert scale ranging from 0 “Not at all” to 6 “Very much” (similar to Charpentier et al., 2018, Vellani et al., 2020, Cogliati Dezza et al., 2022). The dependent variable was the rating of 0-6 . We now changed the text to better clarify that (pg. 5).

Was any questionnaire data collected on individual differences that could be related to information seeking/sharing preferences? For example, positivity/negativity bias or anhedonia?

- 55 participants in Exp1 also performed a set of personality questionnaires (The Apathy Evaluation Scale (AES), The State-Trait Anxiety Inventory (STAI), Revised Life Orientation Test (LOT-R), Zung Self-Rating Depression Scale (SDS)). This served as a pilot for potential future studies. None of these scores were associated with weights assigned to different factors when sharing or seeking information after correcting for multiple comparisons, though given the low N we would not expect significant findings (pg. 6).

Reviewer #3 (Remarks to the Author):

This paper draws a connection between information seeking and information sharing, finding in an online experiment (and a replication thereof) that the propensity to share information is increasing in (prediction) uncertainty, information instrumentality, and positive valence. A key result is that, among those participants who performed both an information seeking and information sharing task (deciding to open or reveal information about their “stock” performance in the experimental “stock market”), there exists a strong correlation between the weight assigned to each factor for both task types (i.e., the relative influence of each factor is stable across information-seeking and information-sharing decisions). This work presents an interesting extension of and contribution to the existing literature. There are, however, several issues that should be addressed, mostly in terms of acknowledging the limitations and making the work more accessible.

- We thank the reviewer for this positive assessment.

With respect to former point, there is no discussion of the work’s limitations. I highlight a few notable ones below, but careful consideration should be given to the possibility of others as well.

External validity and generalizability: As with any experimental setting, the context is sterile and neglects many important features of real-world judgment, decision-making, and behavior. For instance, nothing is known to the focal participant of the other “player” so participants are limited to the three factors of interest (i.e., uncertainty, instrumentality, affective impact) and their own information-seeking preferences to make their information-sharing decision. (The authors somewhat acknowledge this in the line beginning with “Future studies should test the generatability [sic] of these findings...” but it seems a more critical issue and potential alternative explanation for their results than to hand-wave it off as beyond the scope of this research.) Additionally, in the case of stocks, the factors are unambiguous. Would the information value calculation hold in other contexts, for example, socially relevant information?

- Following the reviewer comment we have expanded the discussion to address the limitations of the study and applicability of the results to real life scenarios (pg. 15-16). Specifically, we state “While the three-factor theory of information seeking (Sharot & Sunstein, 2020) has already been demonstrated across different tasks and domains (including financial, social and health decisions: Kelly, C., & Sharot, T. 2021, Molinaro et al., 2023, Cogliati Dezza et al., 2022), this study is the first to test the three-factor theory for information-sharing. If indeed seeking and sharing decisions are closely related we would expect our results to generalize to other domains and to naturalistic tasks. These predictions will need to be tested in future studies. For example, it would be interesting to examine information-sharing in situations where subjects are familiar with the recipients and/or on social media platforms. For example, it is interesting to consider if the three-factor theory could explain the finding that moral outrage leads to more information sharing on Twitter (Brady et al., 2021). It is

possible that users share content that elicits moral outrage because they believe that content is useful in guiding people's action (e.g., has high instrumental utility, for instance by encouraging voting) even if it elicits negative emotions. Alternatively, users may believe the receiver will assess this type of information positively. For example, a statement like 'Trump is a liar' may well elicit a positive response in receivers who are not Trump supporters. That is to say, content that is categorized as eliciting 'moral outrage' may nonetheless be viewed as 'good news'. These different predictions can be tested by probing user's beliefs."

Fatigue/heuristic responding: It seems participants are asked to make a substantial number of decisions across multiple blocks of 40 trials each. Fatigue is always a concern in such situations.

- In response to the reviewer's comment, we investigated the potential influence of fatigue on our findings. To this end, we reanalysed the regression models predicting information seeking and sharing from instrumentality, uncertainty and valence in both Experiment 1 and the replication Experiment. This time, we also included trial number as a control variable. The results revealed that across all conditions, the effects of the three motives remained significant (all $ps < 0.016$), even while controlling for the effects of time (pg. 9).

It is also possible that the 55 and 108 participants (Exp 1 and rep) who completed both tasks established a pattern of responding in the first block that they simply carried over to the second, more for ease of responding than due to a link between their information seeking vs. sharing tendencies.

- We thank the reviewer for this comment. If participants who completed both tasks indeed established a pattern of responding in the first block that they simply carried over to the second, we would expect no difference between the weights assigned to each beta in the sharing and seeking tasks. Independent samples t-tests between weight assigned to each beta in the sharing and seeking tasks (considering both Exp1 and Rep together) suggests this is not the case. Specifically, the instrumental value of information was higher when seeking ($M=0.41$) than when sharing information ($M = 0.28$; $t(132) = 3.36$, $p = 0.001$), indicating that participants did change their response pattern in at least one respect across tasks in a manner that seems consistent with what one would predict. We now report this in the manuscript (pg 9).

Ambiguity: Likely in the interest of avoiding deception, the authors introduce some language that is a bit ambiguous – notably, when they talk about the “recipient” who “may be playing a similar task tomorrow.” This likely introduces some ambiguity and may lead participants to doubt the credibility that there is in fact a real “player” with whom they are sharing the information. This possibility again threatens the authors’ result that individuals project their own information seeking preferences onto others, i.e., if they don’t really think there is an actual “other.”

- The participants did not play with a real recipient, but were led to believe they were doing so. To assess participants' belief in the cover story, we asked 69 participants how they approached the information-sharing task (“How did you decide whether to open the envelope for the other participants?”) and asked two naïve raters to independently rate all responses to indicated whether the responder:

(1) believed they were interacting with another participant. (Options: (a) the responder believes they are interacting with another participant; (b) the responder does not believe they are interacting with another participant; (b) no indication either way).

Analysis of the ratings of the first question revealed that 56.5% of responders explicitly indicated that they believed they were interacting with another participant; 0% of responders explicitly indicated that they did not believe they were interacting with another participant; 43.5% of responses were not informative either way (pg.5-6 in supplementary results).

Task differences: It seems that the two experimental tasks are fundamentally different: “Opening the envelop” in the information-seeking task involves an element of curiosity absent from the information-sharing task, where the participant always has complete information. This seems worth noting in the paper.

- Indeed, a fundamental distinction between seeking and sharing information is that the former involves uncertainty about the information’s content, while the latter assumes full knowledge. This is true in the task as well as in real life, reflecting an inherent characteristic of information seeking and sharing. Importantly, despite this difference, information sharers can still adopt the recipient's perspective and potentially anticipate their curiosity when deciding what to share (pg. 16).

The above issues do not render the work unworthy of publication or, in my opinion, demand additional data, but they do necessitate discussion in the current paper.

- Thank you.

The paper can be made far more accessible with a re-organization and careful attention to gaps in the reader's knowledge – a tricky endeavor when you're so close to the research design but very necessary. The paper would benefit immensely from an upfront roadmap of where the authors are heading, explicit discussion of the specific hypotheses or research goals, and what specific results would be interpreted as support for their predictions. In particular, the finding that individuals prefer to share positive, instrumental, uncertain information is not especially surprising, but the relative weights assigned to each and the relationship between information-seeking vs. -sharing is interesting. It would be good if the authors discussed up front how, specifically, they planned to tease this latter point out given their data.

- Thank you for these suggestions, which we have now incorporated in the revised manuscript. Specifically, we now specify our hypotheses in the introduction (pg. 3-4); (2) explain which data we are going to provide to support our hypotheses in the introduction (pg. 3-4); and (3) provide a road map at the beginning of the Results section (pg. 7).

The section currently labeled “Results” seems actually to be the Methods/Procedure section. Its location is good, and indeed I would highly recommend moving the later Methods/Procedures up, as many of those details are important for the reader to appreciate the methodology (e.g., who are the participants, is decision making incentive compatible, are blocks counterbalanced, etc.).

- Following the reviewer suggestion we moved the “Methods” section before the “Results” one (pg. 4-7).

In particular, participants are told “If they indicated they wanted to open the envelope they were then LIKELY to be shown the current value of their stocks (information).” Upon reading this, one immediately wonders what this means in terms of probability and how this probability relates to participant ratings. Providing all relevant information up-front would be very helpful. (Note: I think the caption for Figure 1 is actually a clearer, more succinct way of articulating what is presented in the five preceding paragraphs. If the authors are concerned about space with relocating the detailed information from the current Methods/Procedure sections to the body of the paper, this is one way of consolidating.)

- Participants indicated their willingness to receive and share information on a 7 Points Likert Scale ranging between 0 and 6, where each point corresponded to a specific probability of receiving the information (5%, 20%, 35%, 50%, 65%, 80%, 95%, respectively) as is past studies (Charpentier et al., 2018, Vellani et al., 2020, Cogliati Dezza et al., 2022). This is now reported on pg 5.

The “Results” section seems actually to begin with the sentence “Participants consider the impact of information on affect, action and uncertainty when deciding whether to inform others” and should thus be labeled as such.

- We have relabelled the section as “Results” (pg 7).

The discussion and interpretation of the (AIC, regression, etc.) results and figures is quite limited and the paper would benefit from a more meaningful discussion thereof.

- We thank the reviewer for this valuable suggestion. We now expand the discussion and interpretation of the regression findings (pg 9), the Linear Mixed Model findings (pg 1, in Supplementary results) and the figures (pg 2 in Supplementary results).

Somewhat relatedly, presenting the raw number of participants from each experiment falling into a given cluster seems less effective than presenting those values as percentages, which would tell the reading something about which cluster is more common.

- We have revised the manuscript to present the percentages of participants within each cluster instead of the raw numbers (pg. 10-11).

The paper has a fair number of typos that should, of course, be corrected (e.g., see [sic] above).

- Thank you for pointing this out. We carefully went over the revised manuscript to correct typos.

Finally, a few points of clarification:

Do analyses include controls for things like block order and variable value for a given trial?

- In our analyses we did not control for block order as order of tasks and blocks was counterbalanced across individuals (pg. 5).

I'm confused by the sentence beginning with “Examining the weight each participant assigned to instrumentality in both tasks revealed that participants who assigned greater weight to a particular factor when seeking information themselves also assigned a greater weight to that factor when deciding whether to share information...” Were some participants not assigned to an instrumentality block? If not, how can you have a coefficient for Instrumentality for those participants? Sorry if I missed something here...

- All participants completed both the instrumental and non-instrumental blocks and for all we compute an instrumentality coefficient. The original sentence refers to the fact that a subset of participants completed both the information-seeking task and the information-sharing task (pg. 5-6).

References

- Brady, W. J., McLoughlin, K., Doan, T. N., & Crockett, M. J. (2021). How social learning amplifies moral outrage expression in online social networks. *Science Advances*, 7(33), eabe5641.
- Caliński, T., & Harabasz, J. (1974). A dendrite method for cluster analysis. *Communications in Statistics-theory and Methods*, 3(1), 1-27.
- Charpentier, C. J., Bromberg-Martin, E. S., & Sharot, T. (2018). Valuation of knowledge and ignorance in mesolimbic reward circuitry. *Proceedings of the National Academy of Sciences*, 115(31), E7255-E7264.
- Krebs, D. (1975). Empathy and altruism. *Journal of Personality and Social psychology*, 32(6), 1134.
- Cogliati Dezza, I., Maher, C., Sharot, T. (2022). People adaptively use information to improve their internal and external states. *Cognition*, 228, 105224.
- Globig, L. K., Witte, K., Feng, G., & Sharot, T. (2021). Under threat, weaker evidence is required to reach undesirable conclusions. *Journal of Neuroscience*, 41(30), 6502-6510.
- Golman, R., Hagmann, D., & Loewenstein, G. (2017). Information avoidance. *Journal of economic literature*, 55(1), 96-135.
- Hertwig, R., & Engel, C. (2016). Homo ignorans: Deliberately choosing not to know. *Perspectives on Psychological Science*, 11(3), 359-372.
- Karlsson, N., Loewenstein, G., & Seppi, D. (2009). The ostrich effect: Selective attention to information. *Journal of Risk and uncertainty*, 38(2), 95-115.
- Kelly, C., & Sharot, T. (2021). Individual differences in information-seeking. *Nature communications*, 12(1), 1-13.
- Krebs, D. (1975). Empathy and altruism. *Journal of Personality and Social psychology*, 32(6), 1134.
- Warneken, F., & Tomasello, M. (2009). The roots of human altruism. *British Journal of Psychology*, 100(3), 455-471.
- Lorah, J., & Womack, A. (2019). Value of sample size for computation of the Bayesian information criterion (BIC) in multilevel modeling. *Behavior research methods*, 51, 440-450.
- Molinaro, G., Cogliati Dezza, I., Bühler, S. K., Moutsiana, C., & Sharot, T. (2023). Multifaceted information-seeking motives in children. *Nature Communications*, 14(1), 5505.
- Raghubir, 2006 Raghubir, P. (2006). An information processing review of the subjective value of money and prices. *Journal of business research*, 59(10-11), 1053-1062.
- Raftery, A. E. (1995). Bayesian model selection in social research. *Sociological methodology*, 111-163.
- Sharot, T., & Sunstein, C. R. (2020). How people decide what they want to know. *Nature Human Behaviour*, 4(1), 14-19.
- Shen, N., & González, B. (2021). Bayesian information criterion for linear mixed-effects models. arXiv preprint arXiv:2104.14725.
- Vellani, V., de Vries, L.P., Gaule, A., Sharot, T. (2020). A selective effect of dopamine on information-seeking. *eLife*, 9, e59152.
- Wagenmakers, E. J. (2007). A practical solution to the pervasive problems of p values. *Psychonomic bulletin & review*, 14(5), 779-804.
- Warneken, F., & Tomasello, M. (2009). Varieties of altruism in children and chimpanzees. *Trends in cognitive sciences*, 13(9), 397-402.

19th Jun 24

Dear Dr Vellani,

Thank you for your patience during the peer-review process. Your manuscript titled "How people decide when to inform others" has now been seen by the same 3 reviewers, and I include their comments at the end of this message. Reviewer 1 has some remaining comments we would like you to address in a revised manuscript before publication. Reviewers 2 and 3 are satisfied with your revision.

We therefore invite you to revise and resubmit your manuscript, along with a point-by-point response to the remaining reviewer. Please highlight all changes in the manuscript text file.

Editorially, we consider it important to add the requested figure and to add statistical robustness checks to address points 1 and 2 or to provide further information if you believe that these are not necessary. We believe the advance of the manuscript is indeed sufficient.

I am attaching an Editorial Requests Table that details critical reporting requirements for the revised manuscript. Please attend to each item and ensure your manuscript is fully compliant. We are requesting that your manuscript aligns with these requirements as this facilitates the evaluation of your manuscript, reducing delays in re-review and potential future acceptance. If your revised manuscript is not aligned with these requests on major issues, such as those concerning statistics, it may be returned to you for further revisions without re-review. Additional information can be found in our style and formatting guide Communications Psychology formatting guide.

Please use the following link to submit your

- revised manuscript,
- point-by-point response to the referees' comments,
- cover letter (as a separate document),
- the Editorial Policy Checklist (see below),
- the Reporting Summary (see below), and
- the completed Editorial Request Table (attached):

[link redacted]

Best regards,

Patricia Lockwood

Patricia Lockwood, PhD

Editorial Board Member

Communications Psychology

orcid.org/0000-0001-7195-9559

REVIEWER REPORTS:

Reviewer #1 (Remarks to the Author):

While I think the authors genuinely engaged in a constructive revision process, and attempted to address reviewers' comments to the best of their abilities, I still have mixed feelings about this manuscript. Some of the raised issues are very satisfactorily addressed and, overall, I find that the manuscript is significantly improved. Yet, I think that some of the main shortcomings of the manuscript persist. Conditional on those limitations, whether this study constitutes the sort of advance appropriate for Communications psychology is a matter of editorial taste (and so I will defer to the editor on that question): I nonetheless list them below, as well as a couple of other comments/suggestions which I hope the authors will find useful to improve the manuscript.

1. I am still somewhat concerned by the limited scope of the findings. The absence of a clear goal for the participants, the deception / potential lack of credibility of the manipulation, the lack of external validity and generalizability (see also R3) still significantly constrain the insights provided by the study. The rebuttal mostly contains a couple of additional, purely descriptive analyses (e.g. about the number of participant that seem to believe the manipulation), but no proper inferential robustness checks (e.g. are the results different if we split believers and non-believers ? If ye split by difference of goals mentioned ?).

2. I am still skeptical about the clustering exercise: The fact that the factors load specifically on the 3 main experimental factors suggests that clustering results are quite idiosyncratic to / driven by the task characteristics, so the insights from those results remain quite limited. Also, given that the data fed to the clustering exercise is the regression betas, shouldn't one also include individual estimates of the regression goodness-of-fit to test whether the clustering is effected by how much the behavior of participants is responsive to the experimental factors, in general ?

3. Despite a couple of requests from another reviewer and myself (which may have been unclear/misunderstood) I am still very surprised by the absence of a main Figure showing the main data/result: i.e. the behavior = $f(\text{factors})$. I would say it is critical that we can get a glimpse at the "raw" Information seeking/sharing levels, as a function of self/other, Instrumentality, valence and uncertainty in a Main Figure. The provided Fig. Supp. 1 does not address this.

Reviewer #2 (Remarks to the Author):

All of my comments have been sufficiently addressed.

Reviewer #3 (Remarks to the Author):

The paper is greatly improved by the revisions, which I feel satisfactorily address the concerns raised in my initial review. I have no additional modifications to suggest.

EDITORIAL POLICIES

We ask that you ensure your manuscript complies with our editorial policies and reporting requirements.

To that end, we require revised manuscripts to be accompanied by two completed items: a reporting summary that collects information on study design and procedure, and an editorial policy checklist that verifies compliance with all required editorial policies.

Nature Research Reporting Summary

Editorial Policy Checklist

All points on the policy checklist must be addressed. Your revised manuscript can only be sent back to the referees if these checklists are completed and uploaded with the revision.

Notes: If you have submitted a Stage 1 Registered Report, Review, Primer, Comment, or Perspective you do not need to submit these forms. If you have already submitted these forms, you may disregard this request.

We thank the reviewers for their guidance and constructive feedback on our manuscript throughout the review process. Their suggestions have greatly enhanced the quality of our work. Below, we address the final few comments from Reviewer 1. For ease of reference, we have included the review below (in BOLD), followed by our response to each concern.

Reviewer 1

While I think the authors genuinely engaged in a constructive revision process, and attempted to address reviewers' comments to the best of their abilities, I still have mixed feelings about this manuscript. Some of the raised issues are very satisfactorily addressed and, overall, I find that the manuscript is significantly improved. Yet, I think that some of the main shortcomings of the manuscript persist. Conditional on those limitations, whether this study constitutes the sort of advance appropriate for Communications psychology is a matter of editorial taste (and so I will defer to the editor on that question): I nonetheless list them below, as well as a couple of other comments/suggestions which I hope the authors will find useful to improve the manuscript.

- We thank the reviewer for their useful comments and suggestions, and we are pleased that they find the manuscript significantly improved. Below we address the remaining comments of the reviewer.

1. I am still somewhat concerned by the limited scope of the findings. The absence of a clear goal for the participants, the deception / potential lack of credibility of the manipulation, the lack of external validity and generalizability (see also R3) still significantly constrain the insights provided by the study. The rebuttal mostly contains a couple of additional, purely descriptive analyses (e.g. about the number of participant that seem to believe the manipulation), but no proper inferential robustness checks (e.g. are the results different if we split believers and non-believers? If we split by difference of goals mentioned?).

- Following the reviewer's suggestion, we compared the betas of participants who explicitly referenced another participant in their debrief and those who did not explicitly referenced another participant. Note, we do not have any participants who explicitly indicated they did not believe there was another participant. There was no difference in the weights the two groups assigned to valence ($t(65) = 1.29, p = 0.2$) nor to uncertainty ($t(65) = -0.07, p = 0.94$). Participants who explicitly expressed belief in the manipulation assigned greater weight to instrumentality upon sharing information ($Mean \beta_{Instrumentality} = 0.28, SE = 0.06$) than those who did not explicitly express such belief ($Mean \beta_{Instrumentality} = 0.03, SE = 0.03$; difference between the two groups: $t(65) = -2.39, p = 0.02$). Similar results were obtained when comparing betas of participants who explicitly expressed their goal as helping others than those that did not: No differences were found across groups for valence betas ($t(65) = 1.53, p = 0.13$) or uncertainty betas ($t(65) = 0.18, p = 0.86$). Participants who explicitly expressed their goal as helping others assigned greater weight to instrumentality upon sharing information ($Mean \beta_{Instrumentality} = 0.28, SE = 0.06$) than those who did not explicitly expressed this goal ($Mean \beta_{Instrumentality} = 0.04, SE = 0.03; t(65) = -3.25, p = 0.002$). We now added these results to the supplementary material.

2. I am still skeptical about the clustering exercise: The fact that the factors load specifically on the 3 main experimental factors suggests that clustering results are quite idiosyncratic to / driven by the task characteristics, so the insights from those results

remain quite limited. Also, given that the data fed to the clustering exercise is the regression betas, shouldn't one also include individual estimates of the regression goodness-of-fit to test whether the clustering is effected by how much the behavior of participants is responsive to the experimental factors, in general?

It appears that we may have not communicated our results clearly. As can be observed in Figure 3, the pattern of results is not such that “the factors load specifically on the 3 main experimental factors”. One of the clusters includes a small significant (or trending) weight on valence along with a medium to large negative weight on uncertainty (which can be viewed as a significant weight on certainty). The other includes a large weight on instrumentality and the third a large weight on uncertainty. We have now highlighted this in the manuscript (pg. 12). We also would like to clarify that cluster analysis does not inherently produce clusters that load on specific factors (like betas 1,2,3) unless the data *naturally* forms such clusters.

3. Despite a couple of requests from another reviewer and myself (which may have been unclear/misunderstood) I am still very surprised by the absence of a main Figure showing the main data/result: i.e. the behavior = f(factors). I would say it is critical that we can get a glimpse at the “raw” Information seeking/sharing levels, as a function of self/other, Instrumentality, valence and uncertainty in a Main Figure. The provided Fig. Supp. 1 does not address this.

- We thank the reviewer for clarifying their request. We have added a new Figure (Fig. 2) to the main text. This figure presents the raw data of information seeking and sharing across all participants and trials (gray points) in both Exp. 1 and the replication, as a function of instrumentality, uncertainty and valence. The black line shows the mean across participants and trials, and the error bars indicating standard error. As can be observed, there is a positive relationship between information seeking/sharing and instrumentality, uncertainty and valence, in both experiments. We believe that this figure directly addresses the reviewer's request for visualization of the raw data.

Reviewer 2

All of my comments have been sufficiently addressed.

- We thank the reviewer for their thorough evaluation of our manuscript and for their helpful suggestions.

Reviewer 3

The paper is greatly improved by the revisions, which I feel satisfactorily address the concerns raised in my initial review. I have no additional modifications to suggest.

- We thank the reviewer for the consideration of the revised manuscript and for the helpful comments. We are pleased to know that the reviewer is satisfied with the revision.

15th Aug 24

Dear Dr Vellani,

Your manuscript titled "How people decide when to inform others" has now been editorially reviewed. I am delighted to say that we are happy, in principle, to publish a suitably revised version in *Communications Psychology*.

We therefore invite you to revise your paper one last time to address the remaining concerns of our reviewers and a list of editorial requests. At the same time we ask that you edit your manuscript to comply with our format requirements and to maximise the accessibility and therefore the impact of your work.

EDITORIAL REQUESTS:

SUBMISSION INFORMATION:

OPEN ACCESS:

* **DATA AVAILABILITY:**

[link redacted]

Best regards,

Jennifer Bellingtier

Jennifer Bellingtier, PhD

Senior Editor

Communications Psychology

Patricia Lockwood, PhD

Editorial Board Member

Communications Psychology

orcid.org/0000-0001-7195-9559